# CODEBENCHGEN: CREATING SCALABLE EXECUTION-BASED CODE GENERATION BENCHMARKS

## ABSTRACT

To adequately test modern code generation systems, evaluation benchmarks must execute and test the code generated by the system. However, these execution and testing requirements have largely limited benchmarks to settings where code is easily executable or has human-written tests. To facilitate evaluation of code generation systems across diverse scenarios, we present CODEBENCHGEN, a framework to create scalable execution-based benchmarks from naturally occurring code sources. Specifically, we leverage a large language model (LLM) to sandbox arbitrary pieces of code into evaluation examples, including test cases for execution-based evaluation. We illustrate the usefulness of our framework by creating a dataset, Exec-CSN, which includes 1,931 examples involving 293 libraries converted from code in 367 GitHub repositories taken from the Code-SearchNet dataset. To demonstrate the solvability of examples in Exec-CSN, we present a human study demonstrating that 81.3% of the examples can be solved by humans and 61% are rated as "requires effort to solve". We conduct code generation experiments on open-source and proprietary models and analyze the performance of both humans and models.[1]

## 1 INTRODUCTION

Code generation systems assist programmers by generating code based on their instructions (Simon, 1963; Feng et al., 2020; Chen et al., 2021). To evaluate the requisite capabilities of such systems, there is a need for benchmarks that simulate a variety of scenarios, such as solving algorithmic problems (Austin et al., 2021; Hendrycks et al., 2021), developing data science applications (Lai et al., 2023), implementing web applications (Oda et al., 2015), and assisting in software engineering practices (Jimenez et al., 2024). Given the complexity of the code that can be generated, it is desirable to execute the system-generated code and check

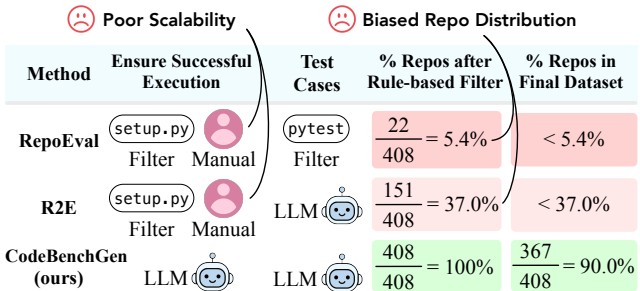

Figure 1: Comparison with existing dataset creation methods.[2] We follow the original paper of RepoEval and R2E and apply their repository filtering strategies on the same repositories we build our dataset from (i.e., 408 repositories in the CodeSearchNet dataset). The final number of repositories in RepoEval and R2E also depend on how much human effort they spend on environment setup, debugging, etc.

whether it passes test cases (Chen et al., 2021; Hendrycks et al., 2021; Lai et al., 2023). Such execution-based metrics have been shown to be a more reliable indicator of functional correctness than execution-free metrics (Chen et al., 2021; Wang et al., 2023; Dibia et al., 2023).

To construct execution-based benchmarks, early works either borrow programming problems from online platforms (Hendrycks et al., 2021; Guo et al., 2024) or manually curate evaluation exam-

---

[1] Code and dataset available at `https://github.com/CodeBenchGen/CodeBenchGen`.
[2] We do not compare to SWE-BENCH (Jimenez et al., 2024) in Figure 1 because they do not introduce their repository selection process.

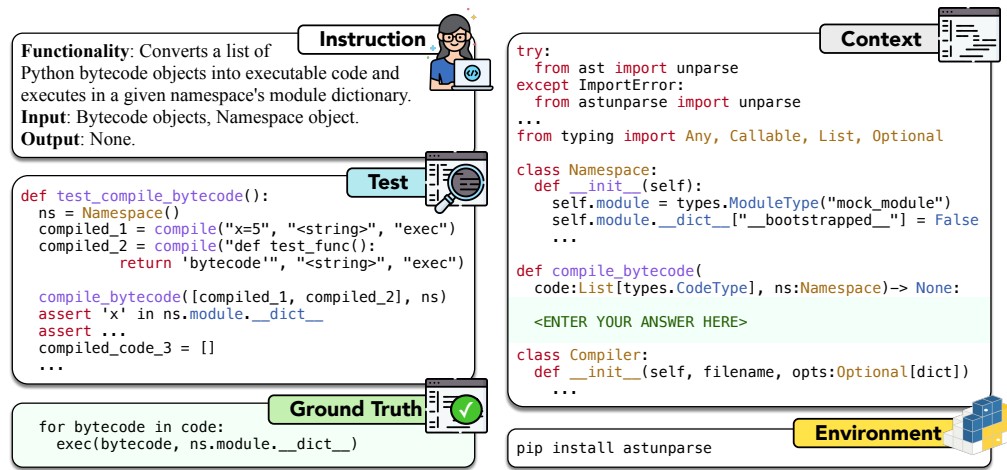

Figure 2: The input of CODEBENCHGEN is an arbitrary code fragment (e.g., code on GitHub), and the output is an evaluation example as illustrated. The context and ground truth are adapted from the input code fragment by an LLM. The instruction and tests are generated based on the adapted code.

ples (Chen et al., 2021; Lai et al., 2023), which are limited to algorithm and data structure problems or require heavy human effort. Recent work builds benchmarks by selecting GitHub repositories containing human-written tests (Zhang et al., 2023; Jimenez et al., 2024), which are generally well-developed projects contributed by professional developers. With a **biased repository distribution**, such datasets cannot directly evaluate models' ability to assist less experienced programmers on less mature projects. A concurrent work (Jain et al., 2024) leverages LLMs to generate tests. However, as shown in Figure 1, to successfully execute the code, it (1) can only build on repositories with setup files, which again limits the created benchmark to mature codebases, and (2) still requires human effort to set up the environment, resulting in **limited scalability**.

In this paper, we present CODEBENCHGEN, a framework to create **scalable** execution-based benchmarks, which leverages an LLM to ensure the successful execution of code. As shown in Figure 2, CODEBENCHGEN adapts an **arbitrary** user-selected code fragment into an evaluation example, where the target is to generate code based on textual instructions and the correctness is checked by test cases. To facilitate the execution of the code fragment, we use an LLM to generate test cases and to *make minimal edits* that allow sandboxing the code. We observe that the LLM can successfully sandbox complicated code such as local module imports, external API usage, and local file reading. To ensure the correctness of the evaluation examples, we iteratively execute and refine the examples until the ground truth code completion can pass all test cases.

As a demonstration of our method's capabilities, we apply the CODEBENCHGEN framework on a set of GitHub code sampled from the CodeSearchNet dataset (Husain et al., 2020) and construct a dataset, Exec-CSN, containing 1,931 examples adapted from 367 GitHub repositories. We successfully adapt 47% of the input code fragments into evaluation examples, covering 90% of the input repositories. Compared to existing datasets, Exec-CSN covers code with more diverse domains (i.e., 293 libraries and 668 repository topics) and repositories created by more diverse programmers (see §4.1 for details). Analyses show that the adapted code preserves high similarity to the naturally occurring input code from GitHub. For instance, the average Jaccard similarity of variables before and after adaptation achieves 83% (§4.2). Although the examples are relatively long, with on average 492 tokens (§4.3), our human study shows that 81.3% of the examples are solvable by computer science graduate students and have a range of difficulties. 85% of the test cases rated as "reasonable and not too trivial", which demonstrates the high quality of our evaluation examples(§4.4).

To evaluate model's ability to solve real-world tasks in diverse domains, we benchmark 12 open-source and proprietary models on Exec-CSN. The key results are as follows: (1) The best model (GPT-4) only achieves a Pass@1 score of 37.21%, indicating that it is still challenging to solve diverse real-world tasks. (2) Specifically, models receive overall lower scores on examples with longer target lengths, more function calls or external libraries, which suggests potential directions to improve current models. (3) We compare the programming abilities of humans and models under a more realistic setting where both humans and models can iteratively improve their answers based on the

execution results. We observe that while GPT-4 has a better initial Pass@1 score, humans achieve significantly better scores after several rounds of improvements.

**Contributions**. (1) We present CODEBENCHGEN, a framework to adapt code fragments into execution-based evaluation examples, which enables researchers to construct scalable and custom benchmarks tailored to naturally occurring code domains. (2) We use our framework to create a new benchmark, Exec-CSN, from GitHub code and conduct analyses and human studies to demonstrate the diversity, realism, complexity, and solvability of the examples. (3) We benchmark 12 code generation models on Exec-CSN and compare and analyze model and human programming abilities.

## 2 METHODOLOGY OF CODEBENCHGEN

The framework of CODEBENCHGEN is illustrated in Figure 4. The input is an arbitrary user-interested code fragment, and the result is an evaluation example for code generation that supports execution-based evaluation, such as illustrated earlier in Figure 2. To construct an easy-to-use benchmark, CODEBENCHGEN provides a single sandbox to run the tests in all the evaluation examples.

**Step 1: Sandboxing**. We need to execute the model-generated code to test generation correctness. However, it is nontrivial to directly execute an arbitrary piece of code from GitHub, especially the code containing complicated dependencies on external files, external API calls, access to the file system, etc.

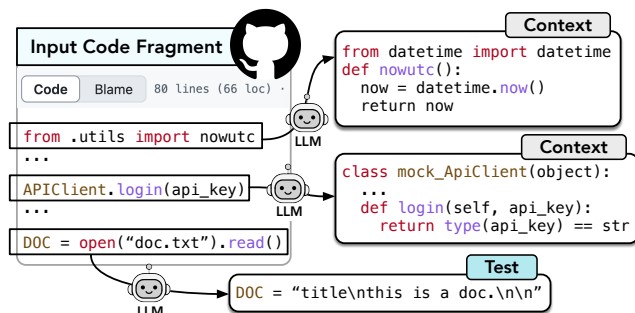

Figure 3: Illustration of the sandboxing step. When adapting the input code, we observe that the LLM can successfully adapt (1) local module imports, (2) external API usage, and (3) local file reading.

As a result, after selecting a segment of the code as the target code, we use an LLM to sandbox the source code so that it can be executed in an isolated environment. To create examples close to real-world code, we prompt the LLM to remove as little source code as much as possible and add new code only when necessary. As shown in Figure 3, in our experiments, we observe that the LLM can (1) generate new classes of functions not presented in the source code, (2) replace external API calls with mock connections, (3) create strings to simulate files in the system, etc.. Furthermore, our quantitative analyses and case studies demonstrate that the sandboxed code is similar to the input code in terms of token-level overlap, variable overlap, and AST depth (See §4.2 for details).

In practice, it is possible that the LLM does not fully follow the instructions and omits too much context or outputs an incomplete piece of code. We thus require the target to be similar to the input and require the context to have a certain length, and automatically re-generate examples that do not satisfy the requirements.

**Step 2: Test Generation**. In the second step, the input is code generated in step 1 and we call an LLM to generate test cases to verify the functionality of the generated code. To encourage test coverage, we prompt the model to generate at least three test cases. Similar to step 1, we check whether the generated tests call the target code and contain at least three assert statements, and re-generate if not.

**Step 3: Iterative Execution and Debugging**. Since the code generated in steps 1 and 2 could have errors, we iteratively execute the code scripts and use an LLM to debug the code until the target output is able to pass all test cases. Unlike previous work that only aims to generate and debug the tests (Xie et al., 2023), we allow the LLM to debug the entire generated code, including the target output, the context, and the tests. Similar to the first two steps, to avoid the adapted code deviating too much from the input, we re-run debugging for the examples where the final generated code has incomplete tests or has a much shorter length than the original input of this step.

**Step 4: Post-processing**. After generating the code, we **generate natural language instructions** for the examples. As discovered by previous work (Wen et al., 2024), adding I/O specifications (i.e.,

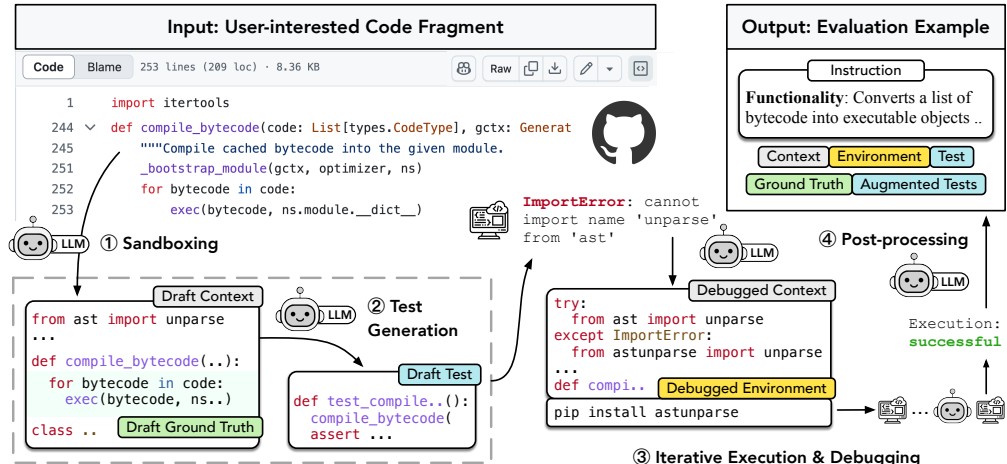

Figure 4: The framework of CODEBENCHGEN, which leverages an LLM to convert a code fragment selected by the user to an evaluation example. The framework (1) sandboxes the code fragment to run in an isolated environment, (2) generates tests for the code, (3) iteratively debugs or regenerates the code to ensure its functional correctness, and (4) post-processes the code into an evaluation example.

the descriptions of the input and output of the target code) better clarifies the intent and consistently improves model performance. We thus use the LLM to generate instructions with natural language descriptions of the desired functionality, input, and output of the target code.

Additionally, we use an LLM to **generate additional tests** (Liu et al., 2023). In evaluation, we execute each set of tests separately and check whether all of them are passed, thereby achieving strictly better test coverage than the initial tests. We note that there are other test generation methods such as fuzzing methods and learning-based methods. As discussed in §6.2, we choose LLM-based methods for its high accuracy.

In previous steps, we sequentially installed the dependencies for each evaluation example, which could potentially conflict with each other. Hence, to set up a shared runtime environment where all of the examples can run without dependency conflicts, we **conduct a final filtering pass**. To do this, we restart the sandbox, install the dependency list, and execute all examples again. The examples with compilation or runtime errors are filtered out. We further filter out examples containing potentially destructive or stateful code. More details on our filtering strategy can be found in Appendix A.2

## 3   CREATING A DATASET: EXEC-CSN USING CODEBENCHGEN

To demonstrate the scalability of our method, we create a dataset, Exec-CSN (examples can be found in Appendix A.7). We take the source code fragment from the CodeSearchNet (Husain et al., 2020) dataset, a GitHub function completion dataset that does not support execution-based evaluation, where the input contains the function signature and the docstring, and the target is the function body. We choose Code-SearchNet as the source because of its large scale and coverage across a diversity of examples from GitHub.

We sample 5,000 Python examples from the test split of CodeSearchNet. Because the function signature only contains limited information, we download the corresponding code file from GitHub as additional

| Step | Description | # Examples |
|------|-------------|-----------|
| ① ② | Sandboxing & Test Gen | 1,260 |
| ③ | 1st Exec & Debug Iter | 1,973 |
| | 2nd Exec & Debug Iter | 2,155 |
| | 3rd Exec & Debug Iter | 2,343 |
| ④ | Post-processing | 1,931 |

Table 1: Number of evaluation examples generated in each step of our framework. We only count examples where the target can pass all the test cases. We have 4,079 pieces of code in the input, covering 408 repositories.

context for each example and filter out examples that do not have a valid file link. To simplify the execution environment of our benchmark, we filter out examples with I/O operations by keywords. There were 4,079 examples left after filtering.

| Dataset | #Examples | #Repo | #Topics | #Libs (Std.+Ext.) | #Contributors | Source |
|---|---|---|---|---|---|---|
| HumanEval (Chen et al., 2021) | 164 | – | – | 4 = 4 + 0 | – | Hand-written |
| MBPP (Austin et al., 2021) | 974 | – | – | 12 = 12 + 0 | – | Hand-written |
| APPS (Hendrycks et al., 2021) | 10,000 | – | – | 0 | – | Coding contest |
| DS1000 (Lai et al., 2023) | 1,000 | – | – | 7 = 0 + 7 | – | Stack Overflow |
| ODEX (Wang et al., 2023) | 945 | – | – | 79 = 48 + 31 | – | Stack Overflow & Hand-written |
| ClassEval (Du et al., 2023) | 100 | – | – | 28 = 19 + 9 | – | Hand-written |
| CoderEval (Zhang et al., 2024) | 230 | 43 | 145 | 179 = 86 + 93 | $1 \sim 417$ | GitHub |
| RepoEval-func (Zhang et al., 2023) | 455 | 8 | 23 | 75 = 33 + 42 | $1 \sim 51$ | GitHub |
| SWE-BENCH (Jimenez et al., 2024) | 2,294 | 12 | 60 | 214 = 111 + 103 | $185 \sim 444$ | GitHub |
| Exec-CSN (ours) | 1,931 | 367 | 668 | 293 = 118 + 175 | $1 \sim 449$ | GitHub |

Table 2: Statistics of Exec-CSN compared to existing execution-based code generation datasets.[3] For datasets built from GitHub, we compare the number of repositories, repository topic, and contributors. We count the libraries in the oracle context for RepoEval-func and SWE-BENCH.

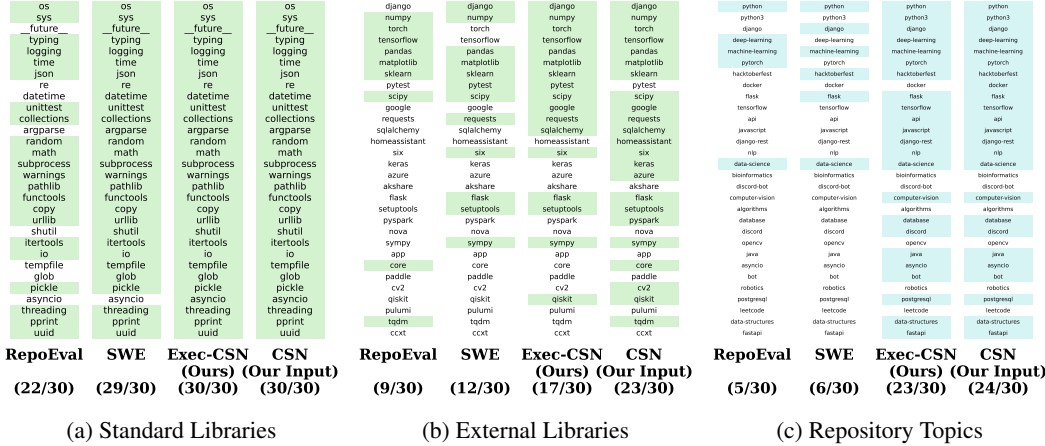

(a) Standard Libraries        (b) External Libraries        (c) Repository Topics

Figure 5: Domain diversity in different datasets. We check each dataset's coverage of the top-30 most common libraries or topics, which are estimated by frequency in the Stack dataset (Kocetkov et al., 2022). "CSN" denotes CodeSearchNet, which we use as input to our framework to create Exec-CSN.

We take the 4,079 examples as the input and run our framework. In the first three steps, we re-generate each example at most three times and run the execution and debugging cycle for at most three iterations per example. We use GPT-4 as the LLM due to its high quality. In post-processing, we use GPT-3.5 to generate 5 more sets of tests for each example and filter out incorrect ones. We use a different LLM in this step to reduce model bias. Details and experiments on test generation can be found in Appendix A.2 and A.3. Table 1 shows the number of evaluation examples generated in each step. After each iteration of debugging, our framework creates more examples where the target can pass all test cases. We finally obtained 1,931 examples after post-processing: successfully converting 47% of the source code fragment into evaluation examples with executable test cases. These 1,931 examples cover 367 out of 408 source repositories from CodeSearchNet, giving 90% repository coverage, which is substantially higher than previous dataset construction methods (Zhang et al., 2023; Jain et al., 2024). Since we take the source code from an existing dataset, the only human effort in constructing Exec-CSN is to make a set of requirements for the examples and write corresponding prompts, which does not grow with the dataset size.

## 4 QUALITY VERIFICATION FOR EXEC-CSN

We evaluate the quality of the generated benchmark by studying three research questions: **[RQ1-Diversity]** Can our framework generate diverse evaluation examples? **[RQ2-Realism]** Are the examples similar to the input code fragment, preserving the realism of code? **[RQ3-Complexity]** Do the examples have varying degrees of complexity? **[RQ4-Solvability]** Are the examples possible to solve, including reasonable instructions, code, and test cases? In this section, we will answer the question with statistics, qualitative analysis, and a human study.

---

[3] We do not compare to R2E-Eval1 (Jain et al., 2024) because their dataset is not released.

Figure 6: Distribution of contributor numbers of the repositories in different datasets.

| | Context + Target | | | Target | | | # Test Cases |
|---|---|---|---|---|---|---|---|
| | #Code Tokens | AST Depth | # Variables | # Code Tokens | AST Depth | # Variables | |
| **Avg** | 491.88 | 9.38 | 12.03 | 86.92 | 7.92 | 4.24 | 8.79 |
| **[min, max]** | [83, 1529] | [6, 18] | [0, 50] | [15, 556] | [4, 18] | [0, 23] | [3, 18] |

Table 3: Complexity of our dataset measured by the number of code tokens, depth of AST, and number of variables in the context and the target code and the number of test cases. We count the code tokens using the `tokenize` library and parse the AST using the `tree-sitter` library.

## 4.1 DIVERSITY ANALYSIS

To answer [RQ1-Diversity], we compare the diversity in (1) the domains of code and (2) the programmers creating the code. Table 2 presents the statistics of Exec-CSN and existing benchmarks *with execution support*. We can observe that Exec-CSN covers code with more diverse libraries (both standard and external), repository topics, and number of contributors.

**Analysis of Domain Diversity**. To estimate the most common domains, we leverage the Stack dataset (Kocetkov et al., 2022), which contains 23M GitHub Python files, and count the libraries and repository topics in 10K randomly sampled files. As shown in Figure 5, Exec-CSN has higher coverage of the top-30 most common standard/external libraries and topics compared to existing execution-based benchmarks. Specifically, RepoEval does not contain `django`, the top-1 external library, and SWE-BENCH misses `torch` and `tensorflow`, which are ranked in the $3^{rd}$ and $4^{th}$ places. In comparison, Exec-CSN covers all top-10 external libraries. We also observe that Exec-CSN covers 30/30 standard libraries, 17/23 external libraries, and 23/24 repository topics from its input code, which suggests that our framework can preserve the diversity in the input to a large extent.

**Analysis of Programmer Diversity**. Figure 6 shows the distribution of the number of contributors to each repository across benchmarks. We observe that compared to existing benchmarks, the distribution of Exec-CSN is closer to the natural distribution evidenced in the Stack. The reason is that RepoEval and SWE-BENCH rely on repositories containing human-written tests, and hence skew toward projects that have high quality, large scale, and more contributors. In comparison, Exec-CSN covers repositories created by diverse contributors, reflecting a wider range of scenarios of coding.

## 4.2 REALISM ANALYSIS

We investigate [RQ2-Realism] by comparing the function to complete in each Exec-CSN example and its corresponding function in the input code (i.e., the GitHub function in the CodeSearchNet dataset). We first compare the BLEU score of the input and adapted function, which is 0.5116 on average, indicating high token-level overlap. As a reference, the BLEU score of two random functions in the input is only 0.0052. We also compare the Jaccard similarity of variables in the input and adapted function. The variable name is an important type of code tokens, which generally suggest the meaning and usage of the variables. As shown in Figure 10, we observe high overlaps between the variables in input and adapted function, with 83% Jaccard similarity on average.

Finally, as shown in Figure 9, we compute the correlation between the number of code tokens, AST layers, and variables in the input and adapted function. We observe high correlation for all three attributes. For instance, the Pearson-$r$ correlation coefficient for number of variables is 0.73, with p-value$< 0.001$. Functions in 62% of the examples have the same AST depth as the input function.

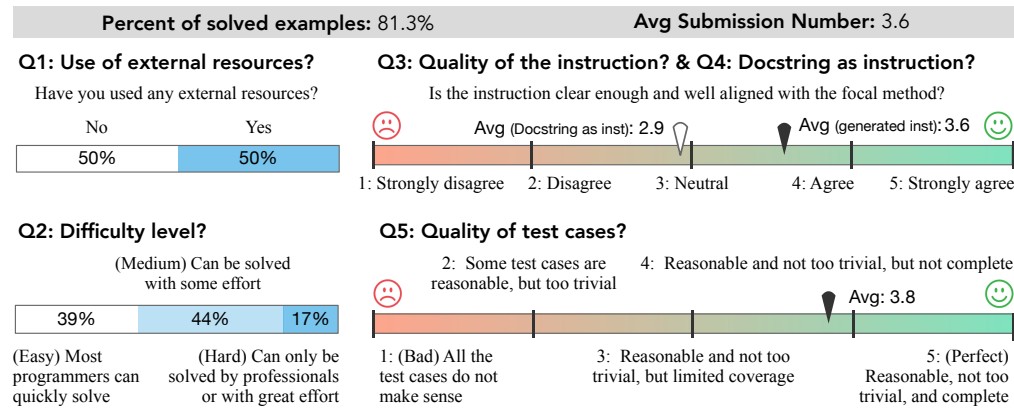

Figure 7: Human study results collected from 64 examples in Exec-CSN.

We conduct a case study in §A.9. In all four cases, the adapted functions are nearly identical to the input. The only minor edit is made in the last case, where our method creates a new class "`DAGS`" to replace an external class "`DagBag`" in the input code.

### 4.3 COMPLEXITY ANALYSIS

To study [RQ3-Complexity], we measure the number of code tokens, depth of AST, and number of variables in both the target and the full example. Results in Table 3 show that our framework can generate relatively long examples, with an average of 491.88 code tokens, 12.03 variables, and 9.38 layers in the AST. The generated examples also have varying complexity, from 83 to 1,529 code tokens. In addition, our framework successfully generates on average 8.79 test cases for each evaluation example, with a high line coverage of 95.76% (See §A.3 for details).

### 4.4 COMPLEXITY AND SOLVABILITY ANALYSIS BY HUMAN STUDY

We conduct a human study to further study [RQ3-Complexity] and [RQ4-Solvability].

**Setup**. We invited computer science graduate student volunteers for the study and obtained results on 64 examples in total. To check the examples' solvability, we present each evaluation example as a coding problem to the participants and ask them to write code based on the instruction and the context. The participants can use any external resources (e.g., search engines) except for AI models. After submitting an answer, they will see the execution results of the test cases and can choose to revise their answers accordingly. We record the percentage of solved problems. Intuitively, a participant solving a problem indicates that the instruction and test cases are consistent with their target code The use of external resources and the number of revisions indicate the complexity level of the examples.

To check the examples' complexity, after the participants pass all test cases or decide not to proceed, we ask them to rate the level of difficulty, the clarity of instructions, and the quality of test cases using a five-point Likert scale. To compare our instruction generation strategy with past work (Husain et al., 2020) that uses the docstrings of the target functions as instructions, we also ask the participants to rate the quality of the docstrings by considering them as instructions.

**Main Results**. Figure 7 presents the human study results. 81.3% of the examples were solved by their assigned participant, which indicates that these examples have clear instructions, reasonable test cases, and have difficulty within a reasonable range. The average rating of our generated instructions is much higher than that of the original docstrings of the target code, which were used as instructions in previous methods (Husain et al., 2020). 85% of the test cases are rated "reasonable and not trivial."

We also observe that the generated examples have varying complexity levels. 34.4% of the examples are solved in the first submission and 68.8% within five submissions. On 50% of the examples, The participants used external resources such as searching for external libraries they are not familiar with. According to the ratings, 39% of the examples can be quickly solved by most programmers. 44% require some effort and 17% can only be solved by professionals or with great effort.

| Model Family | Size | Model | Pass@1 | Pass@2 | Pass@5 | Pass@10 |
|---|---|---|---|---|---|---|
| *Open-source Models* | | | | | | |
| Mistral | 7B | OpenChat-3.5 | 23.61 | 27.08 | 31.39 | 34.34 |
| Llama | 8B | Llama-3-chat | 25.31 | 28.70 | 32.29 | 34.66 |
| CodeQwen | 7B | CodeQwen-chat | 28.58 | 32.54 | 36.82 | 39.49 |
| DeepSeek-Coder | 6.9B | DeepSeek-Coder-chat | 28.68 | 32.36 | 36.52 | 39.22 |
| DeepSeek-Coder | 6.7B | Magicoder-S-DS | 30.73 | 34.92 | 39.56 | 42.47 |
| DeepSeek-Coder | 33B | WizardCoder | 33.35 | 36.81 | 40.51 | 42.76 |
| DeepSeek-Coder | 33B | DeepSeek-Coder-chat | 34.00 | 37.69 | 41.59 | 44.06 |
| CodeLlama | 34B | CodeLlama-chat | 26.24 | 30.11 | 34.37 | 36.95 |
| CodeLlama | 34B | Speechless-CodeLlama | 32.23 | 36.03 | 40.16 | 42.80 |
| CodeLlama | 70B | CodeLlama-chat | 30.92 | 37.36 | **43.60** | **47.18** |
| *Proprietary models* | | | | | | |
| – | – | GPT-3.5 | 32.56 | 35.73 | 39.14 | 41.26 |
| – | – | GPT-4 | **37.21** | **39.85** | 42.67 | 44.53 |

Table 4: Code generation results on Exec-CSN. We put models with different sizes in different groups and split open-source and proprietary models. The model with the best Pass@k performance is highlighted in **bold** and the best open-source model is underlined.

## 5 CODE GENERATION PERFORMANCE EVALUATION

This section aims to answer: [**RQ5**] How do code generation models perform on Exec-CSN?

### 5.1 EXPERIMENTAL SETUP

**Code Generation Models**. We conduct experiments on 10 open-source and 2 proprietary models. We select open-source models across various parameter sizes (7B, 33B, 70B) and model families (Mistral, Llama, CodeQwen, and DeepSeek-Coder). We provide model details in Table 9.

**Evaluation Metrics**. We measure functional correctness using the standard pass@$k$ metric for $k \in \{1, 2, 5, 10\}$ (Chen et al., 2021).

**Experimental Details**. To compute Pass@$k$, we take 20 samples per example for open-source models and 10 samples per example for closed-source models. For all models, we sample outputs with a temperature of 0.3 and top-$p$ of 0.95. Note that we sample with a relatively low temperature as we are primarily interested in Pass@$k$ scores for low $k$; sampling with higher temperature would likely have yielded higher Pass@{5, 10} scores across all models (Chen et al., 2021; Liu et al., 2023). In our prompts, we provide the surrounding context, function header, and docstring, but do not provide the test cases. An example prompt is shown in Appendix A.8.

### 5.2 MAIN RESULTS

Table 4 presents the code generation results of open-source and proprietary models. The best model (GPT-4) only achieves 37.21 Pass@1, indicating that there is still room for improvement on our dataset. Comparison between open-source and proprietary models shows that WizardCoder-33B and DeepSeek-Coder consistently surpass GPT-3.5 on all metrics, although both models are still outperformed by GPT-4. Comparison between models with different sizes shows that both DeepSeek-Coder and CodeLlama greatly benefit from larger parameter sizes.

Results show that CodeLlama-70B achieves the highest Pass@5 and Pass@10 scores, surpassing GPT-4, but does not have a particularly high Pass@1. We observe that although the generated code has high quality, in around 8% of the cases, CodeLlama-70B misinterprets our queries as "dangerous" or as requests to do students' programming homework for them and "refuses" to generate the code, which substantially drag down the Pass@1 score. This is most likely a byproduct of CodeLlama's alignment training, which encourages rejecting dangerous or unethical requests (Rozière et al., 2024).

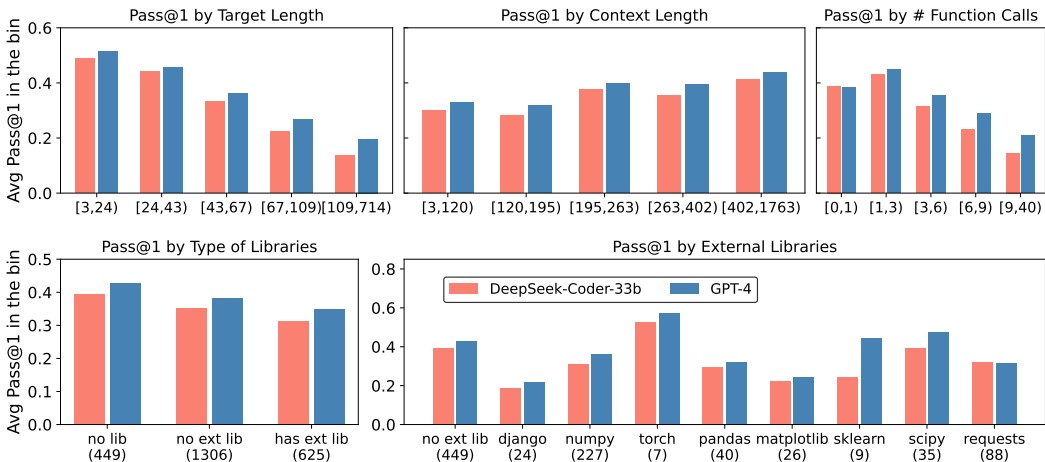

Figure 8: Code generation performance of two models on different groups of examples.

## 5.3 RESULTS ANALYSIS

**Performance Breakdown**. To identify what kind of examples are hard for models, we focus on DeepSeek-Coder-33B and GPT-4, the best open-source and proprietary model under Pass@1, and study four factors that could affect model performance: target length (number of code tokens), context length, number of function calls, and libraries. We split the examples into 5 groups based on each factor such that each group has a similar number of examples. Then we compute the models' average Pass@1 on each group of examples.

As shown in Figure 8, when the target output has a longer length or has more function calls, both models have lower performance and the gap between DeepSeek-Coder and GPT-4 becomes larger. Additionally, both models have lower performance on examples that import any libraries, especially external libraries. However, the context length has a positive but weaker influence on model performance.

We also check the models' performance on examples with different libraries. We inspect the 10 most frequent libraries in the stack, excluding `tensorflow` and `google` as they are present only 4 and 1 times in our dataset, respectively. Results indicate that both the difficulty of generating code and the gap between the two models varies a lot across libraries. For instance, GPT-4 surpasses DeepSeek-Coder substantially on `sklearn` but is slightly outperformed on `requests`. These results motivate benchmarks with high domain diversity to have a more comprehensive ranking of model performance.

| # Revision | GPT-4 | Human |
|---|---|---|
| 0 (Pass@1) | 40.63 | 34.38 |
| 1 | 50.00 | 43.75 |
| 2 | 51.56 | 57.81 |
| 3 | 57.81 | 60.94 |
| 4 | **57.81** | 68.75 |
| 5-17 | – | **81.25** |

Table 5: The performance of humans and GPT-4 on the 64 examples in the human study. We highlight the system with higher and lower accuracy (i.e., percentage of solved examples) after each round of revision. We also **bold** the final accuracy of both systems. The accuracy with no revision is the same as Pass@1.

**Humans vs. Models**. To compare the performance of humans and models, we implement a generation setting similar to the human study that also allows the model to revise from its previous outputs. Specifically, if the model's output fails to pass some test cases, we append its output and the execution results of the test cases to the input and generate again.

We compare human performance to GPT-4 (OpenAI, 2023), the strongest model in our experiments (see §5.2 for details). As shown in Table 5, with no revisions, GPT-4 has a higher Pass@1 than humans. We suspect that the model is trained on code with diverse domains, while human have limited experience in some domains and have trouble solving the problem at the first attempt. However, when revision is allowed, humans can solve much more examples than GPT-4, suggesting that humans make better use of the error messages. Our case study in Appendix A.5 further shows that GPT-4 generates more detailed and complete solutions at the first attempt, while humans write the code with

an incremental approach. The revisions made by humans are also more closely related to the error message, while GPT-4 may output the exactly same wrong answer for multiple times.

# 6 RELATED WORK

## 6.1 CODE GENERATION BENCHMARKS

Researchers have built code generation benchmarks for diverse scenarios, such as query parsing (Zelle & Mooney, 1996), web development (Oda et al., 2015), problem-solving for programmers (Yin et al., 2018), etc. To improve the reliability of evaluation, HumanEval (Chen et al., 2021) evaluates code generation by executing test cases. It is followed by other execution-based benchmarks with larger scale (Hendrycks et al., 2021; Austin et al., 2021), better test coverage (Liu et al., 2023), or multiple languages (Cassano et al., 2022). To extend execution-based evaluation to diverse scenarios, researchers have manually created benchmarks for data science code (Lai et al., 2023), Jupyter notebooks (Yin et al., 2023), user-defined classes (Du et al., 2023), etc.

Recent works further enrich the code generation context by basing benchmarks on repositories with test cases (Zhang et al., 2023; Jimenez et al., 2024). A concurrent work (Jain et al., 2024) uses an LLM to generate test cases instead, but still requires the source repositories to have setup files and need human effort in example curation. Limiting the evaluation to repositories with existing test cases or setup files leads to a biased repository distribution in these datasets, neglecting less mature projects constructed by less experienced programmers. To successfully execute the code, such methods also require human effort to set up the environment, debug the code, or figure out the correct execution commands, which largely limits their scalability. In comparison, our method (1) does not need heavy repository filtering, and (2) uses an LLM to sandbox the code fragments and ensure its successful execution, which has significantly better scalability and repository coverage.

## 6.2 AUTOMATIC TEST GENERATION

Traditional test generation methods include black-box methods that generate random inputs to the code to test (Necula, 2000; Yang et al., 2011; Cha et al., 2015) and white-box methods that analyze the structure of the code (King, 1976; Cadar et al., 2008). Such methods lack context understanding and the generated tests are typically executable but lack relevance. For instance, it may generate a random string as input, while a JSON string is expected. Another category of methods trains language models to generate test cases (Alagarsamy et al., 2023; Tufano et al., 2021), which suffer from unsatisfactory accuracy. Similar to our test generation strategy, recent works prompt LLMs to generate, debug, and improve test cases, which empirically generates more correct unit tests than existing methods (Liu et al., 2022; Xie et al., 2023; Liu et al., 2023). Our work leverages LLM-based test generation, one of the best-performing approaches, as one of the steps of benchmark construction.

# 7 CONCLUSIONS AND FUTURE WORKS

We presented a framework, CODEBENCHGEN, to assist researchers in constructing scalable and custom execution-based code generation benchmarks. The framework leverages LLMs to convert arbitrary code fragments into evaluation examples complete with test cases. We use our framework to create a benchmark, Exec-CSN, with 1,931 examples taken from the CodeSearchNet dataset. We conduct analyses and a human study to verify the quality of generated examples in terms of diversity, realism, complexity, and solvability. Compared to existing code generation benchmarks, Exec-CSN covers more diverse libraries, topics, and number of contributors. We benchmarked open-source and proprietary code generation models on Exec-CSN and analyzed both human and model performance. Results indicate that there is still substantial headroom for models on our dataset.

To improve code generation models, as suggested by our analyses, future work could focus on using more diverse libraries and generating longer code. Based on the comparison with human, another potential direction could be improving the model's ability to iteratively refine the generated code. One may also extend this setting to a more realistic setting, where the model can not only access the execution outputs, but interact with the compiler in a more dynamic way, such as setting break points, printing variable values, or even writing test cases.

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

# A APPENDIX

## A.1 REALISM ANALYSIS: DEVIATION FROM INPUT CODE FRAGMENT

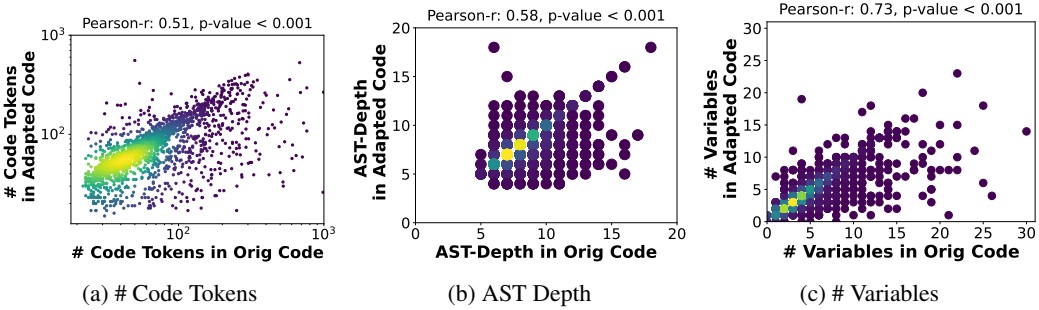

(a) # Code Tokens      (b) AST Depth      (c) # Variables

Figure 9: Comparison between the the statistics of the input code fragment and our adapted code. The statistics include the number of code tokens, depth of AST, and number of variables of the function to complete in Exec-CSN *(y-axis)* and the corresponding function in the input code fragment *(y-axis)*. We report the Pearson-$r$ correlation coefficient and the p-value.

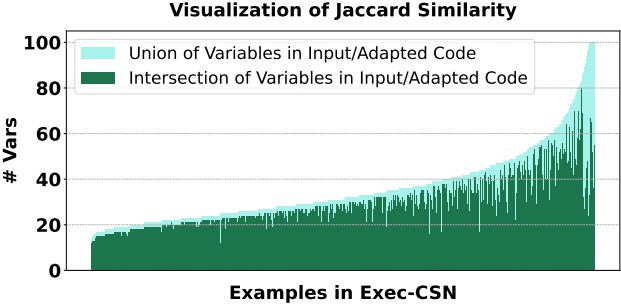

Figure 10: Jaccard similarity of variables in input and adapted functions (83% on average).

## A.2 DATASET POST-PROCESSING DETAILS

In Step 4 of our CodeBenchGen framework (see §2), we perform post-processing on our generated evaluation examples to validate and improve their quality. In this step, in addition to generating natural language instructions with I/O specifications, we filter the data to ensure all examples can run *safely* and generate additional test cases to more accurately measure functional correctness.

**Dataset Post-Filtering**. After all examples have been generated and all dependencies installed, we filter the examples based on two criteria: (1) whether the ground truth code is still able to pass generated tests and (2) whether the example contains potentially dangerous code (e.g., code that kills an OS process).

For the latter, we use simple keyword-based filtering to ban destructive operations, such as killing processes or deleting files. The full list of keywords is shown in Table 6. We find this to be sufficient for our Exec-CSN experiments. However, the keyword list is not complete. An operation not on this list or an external API that invokes one of these operations may still trigger undesirable side effects. Moreover, creating this list of keywords may require some level of domain expertise. We leave the task of creating more secure execution environments to future work.

**Test Augmentation Details**. Execution-based evaluation is heavily dependent on the quality of the executed tests. To improve the reliability of our evaluation, we generate additional tests in post-processing. As explained in §6.2, fuzzing-based methods and learning-based methods for test generation suffer from unsatisfactory relevance or accuracy. As such, we adopt an LLM-based approach similar to that of Liu et al. (2023), where we feed the example being tested to an LLM

```
os.kill                terminate               subprocess.call(['kill',
subprocess.call(['rm', subprocess.call(['rmdir', subprocess.call(["kill",
subprocess.call(["rm", subprocess.call(["rmdir", sys.exit
os.unlink              .unlink                 .rmdir
os.remove              os.removedirs           os.rmdir
os.system              rmtree                  send2trash
open(                  .read                   .write
.load                  .dump                   shutil.
glob.                  os.path.                os.remove(
os.rename(             os.rmdir(               os.mkdir(
os.makedirs(           os.listdir(             .readlines(
.writelines(           .seek(                  .tell(
```

Table 6: List of banned keywords for post-filtering

and prompt it to generate an additional set of tests. To ensure the correctness of generated tests, we discard the sets of tests that the target code cannot successfully pass. To achieve better test coverage, we do not replace the original tests with the generated ones. Instead, we execute the tests separately and the system under test must be able to pass both.

For simplicity, instead of performing iterative execution and self-debugging as in Step 3 of our CODEBENCHGEN framework, we directly sample $k$ tests from the LLM, and select the first of these $k$ tests that is consistent with the target (i.e., the target code passes the tests). If no test is consistent with an example's target code, we do not augment that particular example's tests. In our experiments, we use GPT-3.5 as the test generator as it is relatively inexpensive to sample from while still yielding high-quality outputs. We sample $k = 5$ completions with a temperature of 0.3 and top-$p$ of 0.7.

## A.3    EXPERIMENTS ON TEST AUGMENTATION

**Test Coverage Analyses**. To study the quality of tests augmented by GPT-3.5 in post-processing, we compute the line coverage rate and the performance of models under the original and augmented tests. As shown in Table 7, while the original tests already display high coverage of the target code, they can be further improved by the addition of GPT-3.5-generated tests. Moreover, we find that the augmented tests are able to catch more errors as the Pass@1 scores of our top models drop significantly. Interestingly, however, after adding the new tests, the Pass@1 scores of different models all decrease by roughly the same amount and hence the overall ranking of models remains the same.

To study the test generation ability of different models, we additionally augment the tests with two other models: DeepSeek-Coder-33B and DeepSeek-Coder-7B. As shown in Table 7, combining the tests generated by either model improves the test coverage rate. Among the models, GPT-3.5 generates correct tests for the largest percentage of examples with the highest line coverage rate. Additionally, the coverage of tests generated by DeepSeek-Coder-33B is substantially higher than that of DeepSeek-Coder-7B, suggesting that the test generation ability of the models may also benefit from larger model sizes.

Note that we only keep the tests augmented by GPT-3.5 in Exec-CSN (in addition to the tests generated by GPT-4), as adding tests augmented by DS-33B and DS-7B only leads to marginal improvement in line coverage rate.

**Analyses of Self-bias**. Previous work shows that a wide range of LLMs have self-bias, which means the models rank their own generation output higher than other models' output (Zheng et al., 2023). To study whether the tests generated by LLMs also have self-bias, we check the code generation performance of GPT-3.5 and GPT-4 on the tests generated by GPT-3.5 and GPT-4, separately. We only experiment on the examples where we successfully generate tests by GPT-3.5.

As shown in Table 8, the Pass@1 of GPT-4 is 6.59% higher than GPT-3.5 on GPT-4 generated tests, but is only 3.01% higher on GPT-3.5 generated tests. This suggests that the tests generated by LLMs may also have self-bias. We hence combine the tests generated by the two models, GPT-3.5 and GPT-4, in post-processing to reduce the effects of self-bias.

| Test Cases | % Aug | % Line Coverage | Pass@1 | | | |
|---|---|---|---|---|---|---|
| | | | GPT-4 | DS-33B | GPT-3.5 | DS-7B |
| Original (GPT-4) | - | 94.18 | 42.53 | 38.27 | 37.05 | 33.07 |
| Original + GPT-3.5 | 70.42 | 95.76 | 37.21 | 34.00 | 32.56 | 28.68 |
| Original + DS-33B | 73.33 | 95.32 | 39.79 | 35.77 | 34.66 | 30.54 |
| Original + DS-7B | 52.26 | 95.04 | 41.46 | 37.82 | 36.81 | 32.51 |
| All tests | **88.95** | **96.44** | 33.46 | 30.56 | 30.01 | 25.57 |

Table 7: Line coverage of the target code and downstream Pass@1 performance of the models under the original and augmented tests. The original tests are generated by GPT-4. "% Aug" is the percentage of examples for which we successfully generate additional tests.

| Test Cases | Pass@1 | |
|---|---|---|
| | GPT-4 | GPT-3.5 |
| Tests generated by GPT-4 (original) | 46.28 | 39.69 |
| Tests generated by GPT-3.5 | 41.35 | 38.34 |

Table 8: Performance of GPT-4 and GPT-3.5 under tests generated by GPT-4 or GPT-3.5. Note that all scores are computed over the subset of samples where we have tests generated by both GPT-4 and GPT-3.5.

## A.4 EXPERIMENTAL DETAILS

| Model Family | Size | Model | Full Checkpoint Name |
|---|---|---|---|
| *Open-source Models* | | | |
| Mistral | 7B | OpenChat-3.5 | OpenChat-3.5-0106 (Wang et al., 2024) [4] |
| Llama | 8B | Llama-3-chat | Meta-Llama-3-8B-Instruct (AI@Meta, 2024) [5] |
| CodeQwen | 7B | CodeQwen-chat | CodeQwen1.5-7B-Chat (Bai et al., 2023) [6] |
| DeepSeek-Coder | 6.9B | DeepSeek-Coder-chat | DeepSeek-Coder-7B-instruct-v1.5 (Guo et al., 2024) [7] |
| DeepSeek-Coder | 6.7B | Magicoder-S-DS | Magicoder-S-DS-6.7B (Wei et al., 2023) [8] |
| DeepSeek-Coder | 33B | WizardCoder | WizardCoder-33B-V1.1 (Luo et al., 2023) [9] |
| DeepSeek-Coder | 33B | DeepSeek-Coder-chat | DeepSeek-Coder-33B-instruct (Guo et al., 2024) [10] |
| CodeLlama | 34B | CodeLlama-chat | CodeLlama-34B-Instruct (Rozière et al., 2024) [11] |
| CodeLlama | 34B | Speechless-CodeLlama | speechless-codellama-34b-v2.0 [12] |
| CodeLlama | 70B | CodeLlama-chat | CodeLlama-70B-Instruct (Rozière et al., 2024) [13] |
| *Proprietary models* | | | |
| – | – | GPT-3.5 | gpt-3.5-turbo-0125 [14] |
| – | – | GPT-4 | gpt-4-0125-preview [15] |

Table 9: Evaluated models and their full checkpoint name. We also provide the links to obtain the checkpoints.

---

[4] https://huggingface.co/openchat/openchat-3.5-0106

[5] https://huggingface.co/meta-llama/Meta-Llama-3-8B-Instruct

[6] https://huggingface.co/Qwen/CodeQwen1.5-7B-Chat

[7] https://huggingface.co/deepseek-ai/deepseek-coder-7b-instruct-v1.5

[8] https://huggingface.co/ise-uiuc/Magicoder-S-DS-6.7B

[9] https://huggingface.co/WizardLM/WizardCoder-33B-V1.1

[10] https://huggingface.co/deepseek-ai/deepseek-coder-33b-instruct

[11] https://huggingface.co/codellama/CodeLlama-34b-Instruct-hf

[12] https://huggingface.co/uukuguy/speechless-codellama-34b-v2.0

[13] https://huggingface.co/codellama/CodeLlama-70b-Instruct-hf

[14] https://platform.openai.com/docs/models/gpt-3-5-turbo

[15] https://platform.openai.com/docs/models/gpt-4-and-gpt-4-turbo

### A.5 COMPARING THE CODE REFINEMENT STRATEGY OF HUMANS AND MODELS

To understand why humans have substantially higher performance than models when revision is allowed, we conduct a case study to compare the strategies of humans and models when refining the code they create. Below, we present an example of the evaluation example, the code created by the human in multiple rounds, and the code generated by GPT-4. We have several observations on the difference between human-written and model-generated code.

The first observation is that **models generate more detailed and longer code in the first submission, while humans iteratively add details to the code**. The first few submissions made by humans are often short, with simple code or only calling functions defined in the context ("`self.feed_forward`" and "`accuracy`") in this example. In contrast, models make attempts to cover all problem requirements right from their first submission. Even in cases where the problem description lacks clarity, models make assumptions and confidently output code that seems relevant to solve the problem.

The second observation is that **humans make better use of the error message** when refining unsuccessful submissions. We can see from the example that the refinements made by humans are closely related to the error message. For instance, after seeing the error message about calling the "`feed_forward()`" function, the participant removed the function call from the submission. In contrast, models are often "stubborn," refusing to modify their code even after repeated occurrences of the same error. In the example below, the model keeps generating "`outputs`" and "`targets`" as the input arguments of the "monitors" function, ignoring the error message of "`monitors() missing 2 required positional arguments.`"

The third observation is that **humans make more "careless mistakes" than models**. We observe that human submissions sometimes involve simple syntax errors such as missing the "`:`" after an if statement. In contrast, models rarely make such "careless mistakes".

To sum up, we observe that models typically generate a complete solution in one shot, while humans write code with an incremental approach.

---

**The evaluation example**

**Instructions**

Functionality: This function evaluates and returns the accuracy of model predictions against the set targets using the Classifier's associated loss functions.
Input: An optional dictionary outputs.
Output: Returns a list with one tuple to monitor the accuracy of classifier predictions.

- - - - - - - - - - - - - - - - - - - - - - - - - - - - - - - - - - - - - - - - - - - - - - - - -

**Context**

```python
import numpy as np

class Layer :
    def forward(self, input_data):
        raise NotImplementedError("Forward pass not implemented.")

    def backward(self, grad_output):
        raise NotImplementedError("Backward pass not implemented.")

class Loss :
    def __init__ ( self, targets = {}):
        self.targets = targets

    def accuracy ( self , outputs ) :
        pass

class Regularizer :
    pass

class Classifier :
    DEFAULT_OUTPUT_ACTIVATION = 'softmax'
```

```
    OUTPUT_NDIM = 1

    def __init__ ( self , layers , loss = 'xe' , weighted = False ,
        rng = 13 ) :
        self.layers = layers
        self.losses = [ Loss () ]
        self.weighted = weighted
        self.rng = np.random.default_rng(rng)

    def monitors ( self, outputs = {}, **kwargs) :
        ...

    def feed_forward ( self , x , ** kwargs ) :
        for layer in self.layers:
            x = layer.forward(x)
        return x
```

- - - - - - - - - - - - - - - - - - - - - - - - - - - - - - - - - - - - - - - -

**Target code**

```
    return [('acc', loss.accuracy(outputs)) for loss in self.losses
        ]
```

- - - - - - - - - - - - - - - - - - - - - - - - - - - - - - - - - - - - - - - -

**Test Cases**

```
def test_monitors ( ) :
    model = Classifier ( [ 10 , ( 20 , 'tanh' ) , 50 ] )
    result = model.monitors ( )
    assert isinstance ( result , list ) , 'Result is not a list'
    assert len ( result ) == 1 , 'Result list length is not 1'
    assert result [ 0 ] [ 0 ] == 'acc' , 'First element of result
        tuple is not "acc"'
```

---

### The revision history of human

**Human Submission 1**

```python
def monitors ( self, outputs = {}, **kwargs) :
    acc_list = []
    outputs['predictions'] = self.feed_forward(**kwargs)

    for x in self.losses:
        acc_list.append(x.accuracy(outputs))

    return acc_list
```

**Result**: Error –

```
Traceback (most recent call last):
  File "tmp/split3_test3.py", line 52, in <module>
    test_monitors ( )
  File "tmp/split3_test3.py", line 45, in test_monitors
    result = model.monitors ( )
  File "tmp/split3_test3.py", line 31, in monitors
    outputs['predictions'] = self.feed_forward(**kwargs)
TypeError:feed_forward() missing 1 required positional argument:'x'
```

- - - - - - - - - - - - - - - - - - - - - - - - - - - - - - - - - - - - - - - - - - - - -

**Human Submission 2**

```python
def monitors ( self, outputs = {}, **kwargs) :
    acc_list = []

    for x in self.losses:
        acc_list.append(x.accuracy(outputs))

    return acc_list
```

**Result**: Error –

```
Traceback (most recent call last):
  File "tmp/split3_test3.py", line 52, in <module>
    test_monitors ( )
  File "tmp/split3_test3.py", line 48, in test_monitors
    assert result [ 0 ] [ 0 ] == 'acc' , First element of result
        tuple is not "acc"
TypeError: 'NoneType' object is not subscriptable
```

- - - - - - - - - - - - - - - - - - - - - - - - - - - - - - - - - - - - - - - - - - - - -

**Human Submission 3**

```python
def monitors ( self, outputs = {}, **kwargs) :
    acc_list = []

    for x in self.losses:
        acc_list.append( ('acc', x.accuracy(outputs)) )

    return acc_list
```

**Result**: Successful

---

**The revision history of GPT-4**

**Model Submission 1**

```python
def monitors ( self , outputs , targets ) :
    accuracy_list = [ ]
    for loss in self.losses :
        accuracy = loss.accuracy ( outputs , targets )
        accuracy_list.append ( ( "accuracy" , accuracy ) )
    return accuracy_list
```

**Result**: Error –

```
Traceback (most recent call last):
  File "tmp/code_gen_round0_24.py", line 71, in <module>
    test_monitors()
  File "tmp/code_gen_round0_24.py", line 66, in
test_monitors
    result = model.monitors ( )
TypeError: monitors() missing 2 required positional arguments:
'outputs' and 'targets'
```

- - - - - - - - - - - - - - - - - - - - - - - - - - - - - - - - - - - - - - - -

**Model Submission 2**
Same as Submission 1

- - - - - - - - - - - - - - - - - - - - - - - - - - - - - - - - - - - - - - - -

**Model Submission 3**
Same as Submission 1

- - - - - - - - - - - - - - - - - - - - - - - - - - - - - - - - - - - - - - - -

**Model Submission 4**

```python
def monitors ( self , outputs , targets ) :
    accuracy_list = [ ( loss.__class__.__name__ + "_accuracy" ,
        loss.accuracy ( outputs , targets ) ) for loss in self.
        losses ]
    return accuracy_list
```

**Result**: Error –

```
Traceback (most recent call last):
  File "tmp/code_gen_round4_24.py", line 68, in <module>
    test_monitors()
  File "tmp/code_gen_round4_24.py", line 63, in
test_monitors
    result = model.monitors ( )
TypeError: monitors() missing 2 required positional arguments:
'outputs' and 'targets'
```

A.6   LOGISTIC REGRESSION ANALYSIS ON FACTORS AFFECTING MODEL PERFORMANCE

In addition to the performance breakdown analysis in §5.3, in this section, we present a more rigorous analysis on the factors affecting model performance using logistic regression.

**Analysis Data**. We examine the Pass@1 scores of the following models:

- OpenChat-3.5 (7B)
- DeepSeek-Coder-chat (6.9B)
- Magicoder-S-DS (6.7B)
- WizardCoder (33B)
- DeepSeek-Coder-chat (33B)
- CodeLlama-chat (34B)
- Speechless-CodeLlama (34B)
- CodeLlama-chat (70B)
- GPT-3.5
- GPT-4

We study the performance trends with respect to the following factors, where summary statistics for the above variables are reported in sections 4.1 and 4.3:

- The length of the target code (`TargetLength`)
- The length of the context code (`ContextLength`)
- The number of imports in the code (`NumberImports`)
- The number of variables in the target code (`FocalVariablesCount`)
- The number of variables in the full code (`FullVariablesCount`)
- The AST tree depth for the target code (`FocalASTDepth`)
- The AST tree depth for the full code (`FullASTDepth`)
- The number of calls made to the function in the target code (`NumberFunctionCalls`)

Because the logistic regression analysis requires binary outcome variables, we use the empirical Pass@1 score for this analysis, which is obtained by randomly sampling an output and checking whether or not it passes all test cases. Note that empirical pass@k is different from the standard pass@k metric reported in §5, which is defined as the probability of having at least one correct solution among k randomly selected outputs.

**Analysis Setup**. We performed a logistic regression analysis on examples in Exec-CSN to assess the relative impact each factor had on the empirical Pass@1 score of each model. Specifically, we built a logistic regression model with one instance per example per model with empirical Pass@1 as a categorical dependent measure. After filtering out examples where we encountered errors in AST parsing, we have 1,922 examples left. Thus, there were a total of 19,220 data points in the analysis.

The repositories represent code of different types, thus we included an indicator of repository in our analysis to account for variation across examples that is not represented in any of our co-variates. There were initially 367 distinct repositories, but for the analysis, we aggregated all repositories where we had fewer than 10 examples into a single category called Other. 50 repositories had at least 10 examples. Thus, the `Repository` variable had 51 levels. Independent variables included `Repository`, Model, and the interaction between `Repository` and Model. In addition, we included covariates for Target Length, Context Length, `NumberOfImports`, `FocalVariablesCount`, `FullVariablesCount`, `FocalASTDepth`, `FullASTDepth`, and `NumberFunctionCalls`.

**Analysis Results**. The full model had AIC 22655 and BIC 26699.7. The effect of `Repository` on empirical Pass@1 was significant, indicating that even controlling for the factors represented by the covariates, the examples across repositories were difficult to a differential degree. The category "Other"

was in the middle of the pack. The interaction between `Repository` and Model was not significant, thus indicating that the general performance of models relative to that of the other models did not vary significantly across repositories. The effect of `FullASTDepth` and `NumberFunctionCalls` was not significant. Thus, the interaction term and the two nonsignificant covariates were dropped from the analysis, and the model was rerun.

The full revised model had AIC 218867.7 and BIC 22405.2. All of the independent variables and covariates were significant in the revised model. `TargetLength` and `Repository` had the highest logworth for explaining variation in the dependent variable followed by model, `ContextLength`, `NumberImports`, `FocalVariablesCount`, `FocalASTDepth`, and `FullVariablesCount`. The significant effect of `TargetLength` indicates that a longer `TargetLength` was associated with lower empirical Pass@1. The significant effect of `ContextLength` indicates that lower context lengths were associated with lower empirical Pass@1. The significant effect of `NumberImports` indicates that more imports are associated with lower empirical Pass@1. Higher `FocalVariablesCount`, `FocalASTDepth`, and `FullVariablesCount` were also associated with lower empirical Pass@1.

## A.7 EXEC-CSN DATASET EXAMPLES

Below we show 5 representative examples from Exec-CSN. Note that we only display the original tests for brevity and readability.

In the following example, the code generation model is required to solve a problem using the `re` standard library and a provided regular expression.

---

**Example #1: `files`**

**Instructions**

Functionality: Iterates over text to find & tokenize content within <FILENAME> tags.

Inputs: A single string of text containing <FILENAME> XML-like tags.

Outputs: A generator yielding tuples of `stream_id` and tokenized `tagged_doc` within each <FILENAME> tag.

- - - - - - - - - - - - - - - - - - - - - - - - - - - - - - - - - - - - - - - - -

**File Context**

```python
import re
from nltk.tokenize import WhitespaceTokenizer
filename_re = re.compile ( '''.*?<FILENAME docid="(?P<stream_id
    >.*?)">(?P<tagged_doc>(.|\n)*?)</FILENAME>''' )

def files ( text ) :
    ...
```

- - - - - - - - - - - - - - - - - - - - - - - - - - - - - - - - - - - - - - - - -

**Test Cases**

```python
def test_files ( ) :
    sample_text_1 = '<IGNORE>This is a preamble.<FILENAME docid
        ="12345">Some contents here</FILENAME>And some more text'
    sample_text_2 = 'No filename tag present here at all.'
    sample_text_3 = '<FILENAME docid="67890">Another document
        content</FILENAME><FILENAME docid="54321">Yet another
        document content</FILENAME>'
    results_1 = list ( files ( sample_text_1 ) )
    results_2 = list ( files ( sample_text_2 ) )
    results_3 = list ( files ( sample_text_3 ) )
    assert results_1 == [ ( '12345' , 'Some contents here' ) ] , "
        Test with one FILENAME tag failed."
    assert results_2 == [ ] , "Test with no FILENAME tags failed."
    assert results_3 == [ ( '67890' , 'Another document content' )
        , ( '54321' , 'Yet another document content' ) ] , "Test
        with multiple FILENAME tags failed."
```

- - - - - - - - - - - - - - - - - - - - - - - - - - - - - - - - - - - - - - - - -

**Reference Solution**

```python
def files(text):
    for f_match in filename_re.finditer(text):
        yield f_match.group('stream_id'), f_match.group('tagged_doc
            ')
```

---

This example tests models' ability to use the given context of code to solve a task. In this case, to implement deserialization logic, models must attend to the preceding serialization logic.

---

### Example #2: `Credentials.deserialize`

**Instructions**

Functionality: Reconstructs a Credentials instance from a serialized string.

Inputs: A `serialized_str` representing the serialized credentials string.

Outputs: A Credentials instance with user's protected resources information.

---

**File Context**

```python
import json
from authomatic.exceptions import CredentialsError

class Credentials :

    def __init__ ( self , ** kwargs ) :
        self.token = kwargs.get ( 'token','' )
        self.refresh_token = kwargs.get ( 'refresh_token','' )
        self.expiration_time = int ( kwargs.get ( 'expiration_time'
            , 0 ) )

    def serialize ( self ) :
        concatenated = '\n'.join ( [ self.token , self.
            refresh_token , str ( self.expiration_time ) ] )
        return json.dumps ( { 'credentials' : concatenated } ,
            separators = ( ',' , ':' ) )

    @ classmethod
    def deserialize ( cls , serialized_str ) :
        ...
```

---

**Test Cases**

```python
def test_users ( ) :
    conduit_client = ConduitClient ( )
    assert conduit_client.users ( "PHID-USER-1" ) == json.dumps ( {
        "result" : { "PHID-USER-1" : { "phid" : "PHID-USER-1" , "
        userName" : "user1" } } } )
    assert conduit_client.users ( "PHID-INVALID" ) == json.dumps (
        { "result" : { } } )
    assert conduit_client.users ( "PHID-USER-1" , "PHID-USER-3" )
        == json.dumps ( { "result" : { "PHID-USER-1" : { "phid" : "
        PHID-USER-1" , "userName" : "user1" } , "PHID-USER-3" : { "
        phid" : "PHID-USER-3" , "userName" : "user3" } } } )
```

---

**Reference Solution**

```python
    @classmethod
    def deserialize(cls, serialized_str):
        try:
            # JSON decode.
            data = json.loads(serialized_str)
            token, refresh_token, expiration_time = data['
                credentials'].split('\n')

            # Create a Credentials instance.
            return cls(token=token, refresh_token=refresh_token,
                expiration_time=expiration_time)
```

```
        except Exception as e:
            raise CredentialsError('Failed to deserialize
                credentials: {}'.format(e))
```

This example illustrates how our CODEBENCHGEN framework generates new functions or classes to ensure the self-consistency of the code. In this example, the input code is making HTTP requests, while the information of the website is missing. To avoid making web requests, GPT-4 alters the code to call a mock `request` function which returns a `MockResponse` object. We also observed other workarounds in the data, such as tests that monkey-patch problematic libraries with safer code.

---

**Example #3:** `Client.get_container`

**Instructions**

Functionality: Retrieve information about a specified container, potentially applying filters and settings through query parameters.

Inputs: `container` (required, string), `headers` (optional, dictionary), `prefix` (optional, string), `delimiter` (optional, string), `marker` (optional, string), `end_marker` (optional, string), `limit` (optional, integer), `query` (optional, dictionary), `cdn` (optional, boolean), `decode_json` (optional, boolean).

Outputs: An instance of `MockResponse` containing the status, reason, headers, and contents (either as JSON or a string based on `decode_json`).

- - - - - - - - - - - - - - - - - - - - - - - - - - - - - - - - - - - - - - - - - - - - - - -

**File Context**

```python
import json

class MockResponse :

    def __init__ ( self , status , reason , headers , contents ) :
        self.status = status
        self.reason = reason
        self.headers = headers
        self.contents = contents

class Client :

    def request ( self , method , path , contents , headers ,
        decode_json = False , stream = False , query = None , cdn =
        False ) :
        if method == 'GET' :
            if not cdn :
                mock_contents = json.dumps ( [ { 'name' : '
                    container1' , 'bytes' : 1234 , 'count' : 2 } , {
                     'name' : 'container2' , 'bytes' : 5678 , 'count
                    ' : 5 } , ] )
            else :
                mock_contents = json.dumps ( { "error" : "CDN
                    access not simulated" } )
            return MockResponse ( 200 if not cdn else 400 , "OK" if
                 not cdn else "Bad Request" , { 'content-type' : '
                application/json' } , mock_contents )
        return MockResponse ( 400 , "Bad Request" , { } , "" )

    def get_container ( self , container , headers = None , prefix
        = None , delimiter = None , marker = None , end_marker =
        None , limit = None , query = None , cdn = False ,
        decode_json = True ) :
        ...
```

- - - - - - - - - - - - - - - - - - - - - - - - - - - - - - - - - - - - - - - - - - - - - - -

**Test Cases**

```python
def test_get_container ( ) :
    client = Client ( )
    response = client.get_container ( "my_container" )
    assert response.status == 200
```

```
    assert response.reason == "OK"
    assert isinstance ( response.contents , list ) and response.
        contents [ 0 ] [ 'name' ] == 'container1'
    response = client.get_container ( "my_container" , decode_json
        = False )
    assert isinstance ( response.contents , str )
    response_cdn = client.get_container ( "my_container" , cdn =
        True )
    assert response_cdn.status == 400
```

**Reference Solution**

```
    def get_container(self, container, headers=None, prefix=None,
        delimiter=None, marker=None, end_marker=None, limit=None,
        query=None, cdn=False, decode_json=True):
        query = dict(query or {})
        query['format'] = 'json'
        if prefix:
            query['prefix'] = prefix
        if delimiter:
            query['delimiter'] = delimiter
        if marker:
            query['marker'] = marker
        if end_marker:
            query['end_marker'] = end_marker
        if limit:
            query['limit'] = limit
        response = self.request('GET', '', '', headers, decode_json
            =decode_json, query=query, cdn=cdn)
        if decode_json:
            try:
                response.contents = json.loads(response.contents)
            except json.JSONDecodeError:
                pass
        return response
```

This example tests models' ability to handle longer code contexts as well as their ability to use libraries like `matplotlib`.

---

**Example #4:** `Striplog.plot_axis`

**Instructions**

Functionality: Render a visual representation of geological intervals on a given Matplotlib axis.

Inputs: `ax` (Matplotlib axis), `legend`, `ladder`, `default_width`, `match_only`, `colour`, `colour_function`, `cmap`, `default`, `width_field`, and additional keyword arguments for patch properties.

Outputs: None (modifies the Matplotlib axis in place).

---

**File Context**

```python
import numpy as np
import matplotlib.pyplot as plt
import matplotlib as mpl
from collections import UserDict
from collections import defaultdict

class Position ( UserDict ) :

    def __init__ ( self , z , * args , ** kwargs ) :
        self.z = z
        super ( ).__init__ ( * args , ** kwargs )

class Component ( UserDict ) :

    def __init__ ( self , data = { } , * args , ** kwargs ) :
        super ( ).__init__ ( data , * args , ** kwargs )

class Decor ( UserDict ) :

    def __init__ ( self , width = None , colour = 'black' , * args
        , ** kwargs ) :
        self.width = width
        self.colour = colour
        super ( ).__init__ ( * args , ** kwargs )

class Interval :

    def __init__ ( self , top , base , components = [ ] , primary =
        None , description = '' , data = { } , * args , ** kwargs )
        :
        self.top = Position ( top , data )
        self.base = Position ( base , data )
        self.components = components
        self.primary = primary
        self.description = description
        self.data = data
        super ( ).__init__ ( * args , ** kwargs )

    def thickness ( self ) :
        return self.base.z - self.top.z

class Striplog :

    def __init__ ( self , list_of_Intervals , source = None , order
        = 'auto' ) :
        self._list = list_of_Intervals
        self.order = order
```

```python
        self.source = source
        self.__index = 0

    def plot_axis ( self , ax , legend = None , ladder = False ,
        default_width = 1 , match_only = None , colour = None ,
        colour_function = None , cmap = None , default = None ,
        width_field = None , ** kwargs ) :
        ...

    def get_ylim ( self , ax , z , other = None ) :
        y = z.z
        ymax = 1
        ymin = min ( ymax , y / self.stop )
        return ymin , ymax

    def axis_transform ( self , ax , x , z , data = None , other =
        None , ylim = ( 0 , 1 ) ) :
        return z , x , ylim [ 0 ] , ylim [ 1 ]

    @ property
    def start ( self ) :
        return self._list [ 0 ].top.z

    @ property
    def stop ( self ) :
        return self._list [ - 1 ].base.z
```

**Test Cases**

```python
def test_plot_axis ( ) :
    fig , test_ax = plt.subplots ( )
    test_intervals = [ Interval ( 1 , 2 , components = [ Component
        ( { 'lithology' : 'Limestone' } ) ] , primary = Decor ( 0.5
        , 'gray' ) ) , Interval ( 2 , 3 , components = [ Component (
        { 'lithology' : 'Shale' } ) ] , primary = Decor ( 1.0 , '
        green' ) ) , Interval ( 3 , 5 , components = [ Component ( {
        'lithology' : 'Sandstone' } ) ] , primary = Decor ( 0.75 ,
        'red' ) ) ]
    test_striplog = Striplog ( test_intervals )
    test_striplog.plot_axis ( test_ax , width_field = 'primary' )
    assert len ( test_ax.patches ) == 3 , "There should be 3
        patches on the axis."
    for rect in test_ax.patches :
        assert isinstance ( rect , mpl.patches.Rectangle ) , "Each
            patch should be a Rectangle."
        assert rect.get_width ( ) in [ iv.primary.width for iv in
            test_striplog._list ] , " Rectangle SPACETOKEN width
            SPACETOKEN should SPACETOKEN match SPACETOKEN the
            SPACETOKEN 'primary' SPACETOKEN attribute SPACETOKEN of
            SPACETOKEN the SPACETOKEN intervals."
```

**Reference Solution**

```python
    def plot_axis(self, ax,
                  legend=None,
                  ladder=False,
                  default_width=1,
                  match_only=None,
```

```
            colour=None,
            colour_function=None,
            cmap=None,
            default=None,
            width_field=None,
            **kwargs
            ):
cdata = [getattr(i, width_field) for i in self._list]  #
    Access using single underscore.

for iv, c in zip(self._list, cdata):  # Access using single
     underscore.
    _, ymin = self.get_ylim(ax, iv.base)
    _, ymax = self.get_ylim(ax, iv.top)
    rect = mpl.patches.Rectangle((0, iv.top.z), c.width, iv
        .thickness(), clip_on=False, **kwargs)
    ax.add_patch(rect)
    ax.axvline(c.width, ymin=ymin, ymax=ymax, clip_on=False
        )
```

## A.8 EXEC-CSN CODE GENERATION PROMPT

We provide each model with a prompt consisting of a boilerplate natural language instruction, the surrounding context, and the header and docstring of the target code (which is a function in Exec-CSN). Note that instead of using the original docstring of the function, we use the model-generated functionality-input-output instructions generated in Step 4 (post-processing) of our framework. An example prompt is shown below.

---

**Example Exec-CSN code generation prompt**

Complete the CkClass.flags2text function in the code below based on the docstring. Output one complete piece of code. Your code should start with a ```` ```python ```` delimiter and end with a ```` ``` ```` delimiter.

```python
from __future__ import print_function
import os
import sys

class CkClass ( object ) :
    flags_dict = dict ( )
    fields = dict ( )
    flags = 0

    def flags2text ( self ) :
        """
        Functionality: Converts the 'self.flags' field into a
            list of strings representing set flag bits.
        Inputs: No external inputs; uses class instance's
            'self.flags' and 'self.flags_dict'.
        Outputs: List of strings corresponding to set flags.
        """

        ...
```

## A.9 Original CodeSearchNet Examples vs. Adapted Examples in Exec-CSN

Here we present comparisons between the original CodeSearchNet examples and the adapted sandboxed codes in Exec-CSN. Note that we highlight the focal method in green in the adapted codes. For the majority of examples, the focal method stays (nearly) exactly the same, while surrounding context is added to enable executability.

| **Original CodeSearchNet Function** | **Sandboxed Code** |
|---|---|
| ```python\ndef _psd_mask(x):\n  eigenvalues, _ = tf.linalg.eigh(x)\n  return tf.cast(\n      tf.reduce_min(input_tensor=eigenvalues, axis\n          =-1) >= 0, dtype=x.dtype)\n``` | ```python\nimport numpy as np\nimport tensorflow as tf\n\ndef _psd_mask(x):\n    eigenvalues, _ = tf.linalg.eigh(x)\n    return tf.cast(\n        tf.reduce_min(input_tensor=eigenvalues,\n            axis=-1) >= 0, dtype=x.dtype)\n\ndef test__psd_mask():\n    x1 = np.array([[2, 1], [1, 2]], dtype='float32'\n        ) # Positive Semi-Definite matrix\n    x2 = np.array([[2, -1], [-1, 2]], dtype='\n        float32') # Positive Semi-Definite matrix\n    x3 = np.array([[2, 1], [2, 1]], dtype='float32'\n        ) # Not a Positive Semi-Definite matrix\n    assert tf.equal(_psd_mask(x1), True)\n    assert tf.equal(_psd_mask(x2), True)\n    assert tf.equal(_psd_mask(x3), False)\n``` |
| ```python\ndef iter_labels(self, ontology, size=None,\n    sleep=None):\n    for label in _help_iterate_labels(self.\n        iter_terms(ontology=ontology, size=\n        size, sleep=sleep)):\n        yield label\n``` | ```python\nimport logging\nimport time\nimport requests\nfrom unittest.mock import patch, Mock\n\nHIERARCHICAL_CHILDREN = 'hierarchicalChildren'\napi_ontology = '/api/ontologies/{ontology}'\napi_terms = '/api/ontologies/{ontology}/terms'\n\ndef _iterate_response_terms(response):\n    for term in response['_embedded']['terms']:\n        yield term\n\ndef _help_iterate_labels(term_iterator):\n    for term in term_iterator:\n        yield term['label']\n\nclass OlsClient:\n    """Wraps the functions to query the Ontology\n        Lookup Service."""\n    def __init__(self, ols_base):\n        {function body omitted for brevity}\n    def iter_terms(self, ontology, size=None, sleep\n        =None):\n        {function body omitted for brevity}\n\n    def iter_labels(self, ontology, size=None,\n        sleep=None):\n        for label in _help_iterate_labels(self.\n            iter_terms(ontology=ontology, size=\n            size, sleep=sleep)):\n            yield label\n\ndef test_iter_labels():\n    with patch('requests.get') as mocked_get:\n        mocked_get.return_value.json.return_value =\n            {\n            '_embedded': {\n                'terms': [{'label': 'term1'}, {'\n                    label': 'term2'}, {'label': '\n                    term3'}]\n            }\n        }\n        ols_client = OlsClient(ols_base='\n            ols_api_base')\n        labels = list(ols_client.iter_labels(\n            ontology='ontology_name'))\n        assert labels == ['term1','term2','term3']\n        assert len(labels) == 3\n        assert 'term2' in labels\n``` |

| Original CodeSearchNet Function | Sandboxed Code |
|---|---|

```python
def _connect(self):
    with self._lock:
        if self._aggregator:
            try:
                return self._pool_connect(self.
                    _aggregator)
            except PoolConnectionException:
                self._aggregator = None

        if not len(self._aggregators):
            with self._pool_connect(self.
                    _primary_aggregator) as conn:
                self._update_aggregator_list(
                    conn)
                conn.expire()

        random.shuffle(self._aggregators)

        last_exception = None
        for aggregator in self._aggregators:
            self.logger.debug('Attempting
                connection with %s:%s' % (
                aggregator[0], aggregator[1])
                )

            try:
                conn = self._pool_connect(
                    aggregator)
                # connection successful!
                self._aggregator = aggregator
                return conn
            except PoolConnectionException as e
                    :
                # connection error
                last_exception = e
        else:
            # bad news bears...  try again
                later
            self._aggregator = None
            self._aggregators = []

        raise last_exception
```

```python
import random
import threading
import logging

class DummyError(Exception):
    pass

class ConnectionPool:
    {class contents omitted for brevity}

class PoolConnectionException(Exception):
    pass

class conn:
    {class body omitted for brevity}

class RandomAggregatorPool(object):
    def __init__(self, host, port, user='root',
            password='', database='information_schema
            '):
        {function body omitted for brevity}

    def connect(self):
        {function body omitted for brevity}

    def connect_master(self):
        {function body omitted for brevity}

    def close(self):
        self._pool.close()

    def _pool_connect(self, agg):
        return self._pool.connect(agg[0], agg[1],
            self._user, self._password, self.
            _database)

    def _connect(self):
        with self._lock:
            if self._aggregator:
                try:
                    return self._pool_connect(self.
                        _aggregator)
                except PoolConnectionException:
                    self._aggregator = None

            if not len(self._aggregators):
                with self._pool_connect(self.
                        _primary_aggregator) as conn:
                    self._update_aggregator_list(
                        conn)
                    conn.expire()

            random.shuffle(self._aggregators)

            last_exception = None
            for aggregator in self._aggregators:
                self.logger.debug('Attempting
                    connection with %s:%s' % (
                    aggregator[0], aggregator[1])
                    )

                try:
                    conn = self._pool_connect(
                        aggregator)
                    self._aggregator = aggregator
                    return conn
                except PoolConnectionException as e
                        :
                    last_exception = e
            else:
                self._aggregator = None
                self._aggregators = []
                raise last_exception

    def _update_aggregator_list(self, conn):
        {function body omitted for brevity}

    def _refresh_aggregator_list(self, conn):
        {function body omitted for brevity}
```

| Original CodeSearchNet Function | Sandboxed Code |
|---|---|

```python
def get_dag_run_state(dag_id, execution_date):
    """Return the task object identified by the
        given dag_id and task_id."""

    dagbag = DagBag()

    # Check DAG exists.
    if dag_id not in dagbag.dags:
        error_message = "Dag id {} not found".
            format(dag_id)
        raise DagNotFound(error_message)

    # Get DAG object and check Task Exists
    dag = dagbag.get_dag(dag_id)

    # Get DagRun object and check that it exists
    dagrun = dag.get_dagrun(execution_date=
        execution_date)
    if not dagrun:
        error_message = ('Dag Run for date {} not
            found in dag {}'
                            .format(execution_date,
                                dag_id))
        raise DagRunNotFound(error_message)

    return {'state': dagrun.get_state()}
```

```python
import random
from datetime import datetime

class DagNotFound(Exception):
    pass

class DagRunNotFound(Exception):
    pass

class DummyDag:
    def __init__(self, dag_id):
        self.dag_id = dag_id

    def get_dagrun(self, execution_date):
        if execution_date.year == 2020:
            return DummyDagRun()
        else:
            return None

class DummyDagRun:
    def get_state(self):
        return random.choice(['success', 'failed',
            'running'])

DAGS = {
    'example_dag_1': DummyDag('example_dag_1'),
    'example_dag_2': DummyDag('example_dag_2'),
    'example_dag_3': DummyDag('example_dag_3'),
}

def get_dag_run_state(dag_id, execution_date):
    """Return the task object identified by the
        given dag_id and task_id."""
    if dag_id not in DAGS:
        error_message = "Dag id {} not found".
            format(dag_id)
        raise DagNotFound(error_message)

    dag = DAGS[dag_id]
    dagrun = dag.get_dagrun(execution_date=
        execution_date)
    if not dagrun:
        error_message = ('Dag Run for date {} not
            found in dag {}'
                            .format(execution_date,
                                dag_id))
        raise DagRunNotFound(error_message)

    return {'state': dagrun.get_state()}
```

