# OpenReview forum: "CodeBenchGen: Creating Scalable Execution-based Code Generation Benchmarks"
_ICLR.cc/2025/Conference — Submitted to ICLR 2025_

### Official Review · Reviewer_Qpht · 2024-11-03

**Soundness:** 3
**Presentation:** 3
**Contribution:** 1
**Rating:** 3
**Confidence:** 5

**Summary:**

This paper introduces CODEBENCHGEN, a framework designed to create scalable execution-based benchmarks for code generation systems, leveraging a LLM to convert code fragments into evaluation examples. The resulting dataset, Exec-CSN, derived from GitHub code, aims to provide a benchmark that addresses limitations in existing datasets, such as biased repository selection and lack of scalability. The paper presents thorough evaluations, including human and model performance on Exec-CSN.

However, the dataset is built entirely on the performance of LLM, and the method is still limited to simple model calls. I noticed that many more detailed works generate test cases for repo-level code by enhancing LLM. This makes it seem that this article is too simple in its contribution, except for the discussion of model bias cases, to be less than an evaluation benchmark - because the data itself has many biases and inherent problems in the SE field based on the model, such as data leakage and false code coverage. I noticed that the author did not seem to pay attention to the repo-level code generation benchmarks recently published in conferences such as ACL and Neurpls, such as EvoCodeBench. We generally believe that there is more abundant executable test case information in the tests that come with mature projects, which seems to be more worthy of testing and trust than LLM generation.

In general, I think the focus of this article is very confusing. It wants to expand the coverage and quantity of the data itself, but there is no innovative method to solve the problem of LLM's inherent normalization of repo-level code data. In the end, it proposes a very strange benchmark that can be considered as a simple call to LLM to generate some targeted test cases for a wide range of code repositories.

**Strengths:**

1. **Good Motivation**: The authors leverage LLMs to address execution-based benchmarks by creating scalable benchmarks from naturally occurring code sources. This approach overcomes manual filtering constraints and scales to diverse domains.

2. **Dataset Diversity and Realism**: Exec-CSN presents an improvement over previous benchmarks in scope. The authors provide a detailed comparison showing that Exec-CSN achieves high domain diversity and is applicable to a broader range of programming scenarios, which is crucial for evaluating generalizable code generation systems.

3. **Evaluation of Human and Model Performance**: The experiments involving multiple open-source and proprietary models (such as GPT-4) on Exec-CSN reveal valuable insights into the strengths and limitations of current models. Additionally, the human study adds credibility to the dataset's difficulty distribution and solvability, ensuring that the examples reflect real-world tasks with a reasonable complexity spectrum.

**Weaknesses:**

1. **Technical Pipeline**: The multi-step pipeline (sandboxing, test generation, debugging, and post-processing) is well-documented but lacks sufficient detail in certain stages. Specifically, the role of LLM in sandboxing and iterative debugging could be clearer, especially regarding how errors in test generation or sandboxing are handled.

Moreover, this article proposes that LLM has a poor accuracy rate when revising and debugging complex code projects, but in this part, the model is still used for high-degree-of-freedom debugging. I think there is a serious risk of reducing code complexity.

2. **Comparison with Related Benchmarks**: While the paper highlights the limitations of current benchmarks, it would benefit from a more in-depth discussion on the specific aspects that Exec-CSN improves upon compared to recent LLM-generated benchmarks. More comparative analysis on dataset quality and code realism across these benchmarks would further support Exec-CSN's claimed advantages.

3. **Scalability and Environment Constraints**: The paper briefly mentions that a final filtering pass is applied to ensure examples are executable without dependency conflicts, but more specifics about the limitations this filtering imposes would be helpful. For instance, how does the final dataset address examples requiring unique or uncommon dependencies?

4. **Dependent on model performance**: The key of this dataset——executable test cases is built entirely on the performance of LLM, and the method is still limited to simple calls. This makes this article simple in its contribution, also to be less than an evaluation benchmark - because the data itself has many biases and inherent problems in the SE field based on LLM. The author mentioned the model's bias towards self-generated test cases, but further analysis of the causes and solutions to the bias is obviously more important than proposing a benchmark with flawed test cases.

In general, I think the focus of this article is very confusing. It wants to expand the coverage and quantity of the data itself, but there is no innovative method to solve the problem of LLM's inherent normalization of repo-level code data. In the end, it proposes a very strange benchmark that can be considered as a simple call to LLM to generate some targeted test cases for a wide range of code repositories.

**Questions:**

1. **Provide Example Case Studies**: Adding detailed examples of code fragments before and after adaptation by CODEBENCHGEN, alongside generated tests, would help readers better understand the transformations applied and the framework's overall impact on code realism.

2. **Expand Discussion on Solvability and Usability**: The human study highlights solvability but should delve deeper into usability aspects, such as whether participants found the instructions or debugging aids intuitive. Addressing user feedback on example usability could provide insights for refining the dataset.

---

> ### Author Response · Authors · 2024-11-27
>
> Thanks for your feedback. In the response, we will first introduce the motivation of our work.
>
> Regarding your concern about high-degree-of-freedom debugging and the realism of LLM-generated code, **we add a quality check step in our framework and filter out failed examples (see part 1 of the General Response for details)**.
>
> Regarding the comparison with existing frameworks, **we apply R2E and CodeBenchGen to construct datasets based on the same 50 recent repositories** . Results show that even with the above quality filtering step, our method still has a much higher success rate and repository coverage than R2E, a recent benchmark construction work (see part 2 of the General Response for details).
>
> We will then answer your other questions.
>
> &nbsp;
> ## Motivation
>
> There are two major motivations to have a scalable method to create code generation benchmarks:
> (1) **Reduce data contamination**: with the development of new and improved LLMs, there’s always a risk that the LLMs are already trained on the evaluation benchmarks (as shown by previous work: https://arxiv.org/pdf/2403.07974). If we have a scalable method, we can easily construct benchmarks based on newly-created codebases that are released after the LLM’s checkpoint, and hence reduce data contamination.
> (2) **Cater to user’s interest**: Although there already exists some code generation benchmarks, the users may be interested in some specific domains that are not covered by any existing benchmarks. With a scalable dataset creation method, the users can simply find the repositories of some specific topic, and quickly build a dataset on them.
>
> Most existing code generation benchmarks either rely on human-written setup files or tests (e.g., RepoCoder, SWE-Bench). R2E extracts GitHub functions that can be executed with no bugs and uses LLM to generate test cases. In comparison, we take a step forward and use LLMs to (1) make an arbitrary code fragment executable, (2) generate test cases, and (3) generate instructions for code generation. Our framework requires less human effort and is more scalable. Our experiments also show that our method has higher success rates than R2E (i.e., **Compared to R2E, CodeBenchGen can convert a wider range of code fragments into high-quality evaluation examples**).
>
>
>
> &nbsp;
> ## W 1 & 4: Quality check for sandboxing and debugging & Comparison to the real GitHub Code
>
> As described in part 1 of the General Response:
> * We use AST-based comparison and LLMs (GPT-4o) to check (1) the functionality equivalence with the seed code, (2) test correctness, and (3) instruction clarity. Then we filter out examples that fail on any condition.
> * We conducted a human study to show that the LLM verifier has a high agreement with human.
> * Results show that 55.83% of the Exec-CSN examples pass all the verifications. Even if we filter out other examples, they still have a wide coverage (307 repositories) and keep the similar complexity as before.
>
> By only keeping the examples where the target function has the same functionality as the seed function, **we ensure that we do not reduce its complexity or realism during sandboxing or debugging**. We hope this resolves your concern about “reducing code complexity” and “the data’s biases and inherent problems based on LLM”.
>
>
>
>
> &nbsp;
> ## W2: Comparison to R2E
>
> In short, starting from the same set of repositories, we apply R2E and CodeBenchGen to construct datasets. We show that our method is able to construct much more examples than R2E (196 vs. 17 examples). With the above quality filtering strategy, **similar to R2E, we can also ensure that the functionality of the target function is the same as the seed code, keeping the realism of the examples**. We hope this can resolve your concern about the comparison with existing benchmarks.
>
> Specifically, here’s the summary of the experimental details (part 2 of the General Response):
> * We start with the same set of repositories (50 randomly selected licensed repositories with recent updates) and apply R2e and CodeBenchGen to produce datasets
> * R2E filters our repositories without setup files and functions without docstrings, and CodeBenchGen does not need those filterings, resulting in a larger number of seed functions.
> * CodeBenchGen is able to debug the environment errors or other execution errors in the repositories. As a result, CodeBenchGen produces a much larger final dataset (with 196 examples from 29 repositories), while R2E is only able to produce 17 examples from 3 repositories.

---

> > ### Author Response · Authors · 2024-11-27
> >
> > &nbsp;
> > ## Q1: Case studies
> > We provide a case study in Appendix A.9, which shows the original GitHub function and our evaluation example based on it. We highlight several modifications made by our method during the sandboxing stage: (1) generate new classes of functions not presented in the source code, (2) replace external API calls with mock connections, (3) create strings to simulate files in the system, etc..
> >
> > In the next version of the paper, we will move at least one case study example to the main context and add more case studies to the Appendix. We will also include the above quality check & filtering strategy and the human study results.

---

### Official Review · Reviewer_wxVu · 2024-11-03

**Soundness:** 2
**Presentation:** 3
**Contribution:** 3
**Rating:** 3
**Confidence:** 5

**Summary:**

This paper presents CodeBenchGen, an automated framework that generates code generation benchmarks from arbitrary code fragments. Given a code fragment, CodeBenchGen uses GPT-4 to rewrite it to make it executable, e.g., adding missing import statements, adding function/class headers, etc. It then prompts GPT-3.5 to generate test cases for the rewritten code. The authors chose two different LLMs for code rewriting and test generation to avoid model bias. Given that GPT-3.5 may generate test cases that are not aligned with the rewritten code, CodeBenchGen further debugs and repairs the rewritten code iteratively until it passes all test cases. CodeBenchGen also continues to prompt GPT-3.5 to generate a natural language description for the code and additional test cases. The authors evaluated CodeBenchGen on 4079 code fragments sampled from CodeSearchNet. CodeBenchGen successfully converted 1931 (47%) of them to code generation tasks with test cases. These tasks cover more libraries and are originally from a more diverse set of GitHub repositories compared with existing code generation benchmarks. The authors further did a human study and found that 81% of these tasks were considered solvable by CS students. Finally, the authors tested 12 LLMs on this benchmark and found that the best model (GPT-4) only achieved 37.21% pass@1 on this benchmark.

**Strengths:**

1. This work addresses an interesting and important problem in LLM-based code generation. There have been a lot of concerns about existing benchmarks like data leakage and tasks that are too simple. The proposed framework has good potential to address these concerns and continually generate new benchmarks automatically.

2. The proposed framework sorely relies on LLMs to convert code fragments to code generation tasks with test cases. Prior work like R2E (NeurIPS 2024) and EvalPlus (NeurIPS 2023) use LLMs and conventional program analysis methods like program slicing and mutation analysis. Thus, only replying on LLMs seems to be a technical novelty of this work.

3. The authors did a human study to evaluate the quality of code generation tasks. This kind of qualitative analysis looks novel and provides interesting insights.

**Weaknesses:**

While I find this work interesting and has good potential, it still requires a lot of improvement in terms of technical details, rigor, and evaluation.

1. My first concern is that the prompts used to rewrite code, generate test cases and debug and repair code are not provided at all. Given that the proposed framework is essentially an LLM pipeline and LLMs are sensitive to the prompt design, the authors should provide the prompts and elaborate on the prompt design. Without seeing the prompts, it is hard to assess the technical soundness of this work. This also leads to a major reproducibility issue for other researchers in the future.

2. There is no evaluation of the individual steps of the benchmark generation framework. Given that the intellectual merit of this work is the automated benchmark generation framework, it is important to provide a thorough evaluation of the framework's individual steps. However, the current evaluation only focuses on measuring the diversity and quality of the benchmark generated from CodeSearchNet. While it is okay to demonstrate the generated benchmark is more diverse, only focusing on this misses a more critical evaluation aspect of the framework itself. Besides, the benchmark has limited utility in practice since CodeSearchNet has been used as the pre-training dataset for many code LLMs.

3. Compared with other similar frameworks like R2E and EvalPlus, the novelty of the proposed framework is that it only relies on LLMs to generate code generation tasks. However, there is no evaluation to demonstrate the benefit of sorely relying on LLMs. The authors should demonstrate that this proposed approach is better than R2E and EvalPlus. For example, the authors may want to show that CodeBenchGen has a higher success rate than R2E and EvalPlus on the same set of code fragments. They may also want to show that CodeBenchGen can handle more complex data types or complex code fragments. The authors said R2E's code was not available, but I found it here https://github.com/r2e-project/r2e

4. A downside of only replying on LLMs is the cost of repeatedly prompting LLMs, especially given that CodeBenchGen needs to iteratively prompt GPT-4 to debug and repair the code to pass all test cases. The authors should report how many prompts were issued to GPT-4 and GPT-3.5 for each task and what the cost is.

5. The human study lacks details and also has several rigor issues. This paper doesn't mention how many participants were recruited and how the 64 code generation tasks were selected. It is also unclear how the tasks were assigned to the participants. Was each task evaluated by multiple participants? If so, what is the agreement level?

6. The case study in A.9 provides interesting insights into the code rewriting step in the automated framework. However, it is only based on four examples. As I mentioned earlier, the intellectual merit of this work is the automated framework. Expanding this case study and bringing it to main text could be an effective way to address my previous concern. While I understand there is a page limit, the authors could consider moving some less solid or less interesting experiments to the appendix to make space for this case study. For example, the analysis of programmer diversity is not a solid experiment since there is no established evidence in the literature that the number of contributors/programmers is positively correlated with the quality of code generation tasks.

7. Both R2E and EvalPlus have specifically designed mechanisms to make sure they do not alter the semantics/functionality of the original code fragment. They iterate on the test outputs to make sure the test cases match the semantics/functionality of the original code fragment. However, this work adopts an opposite design. CodeBenchGen keeps debugging and repairing the original code fragment until it passes the test cases generated by LLMs. This sounds counter-intuitive since this process is likely to change the functionality of the original code. While the authors measure the similarity between the original code and the final code using AST similarity and token similarity, those metrics do not capture the semantic similarity between the original code and the final code.

**Questions:**

In addition to the concerns above, I also have a question about the claim on R2E. While I understand that R2E requires the existence of setup files to install dependencies and configure the testing environment, I don't quite get what kinds of manual effort are required in R2E. Can you elaborate a bit on this?

---

> ### Author Response · Authors · 2024-11-27
>
> Thanks for your interest in our work and your valuable feedback!
>
> In summary, regarding your concern about example quality, **we add a quality check step in our framework and filter out failed examples (see part 1 of the General Response for details)**.
>
> We also take your advice and have a comparison with R2E. **Starting with the same set of repositories, we show that our method has a much higher success rate and repository coverage than R2E (see part 2 of the General Response for details)**.
>
> We also provide the answer to your other questions.
>
> &nbsp;
> ## W2 & 6 & 7: Evaluation of individual steps & Quality check mechanisms & Case studies
>
> As described in part 1 of the general response:
> * We use AST-based comparison and LLMs (GPT-4o) to check (1) the functionality equivalence with the seed code, (2) test correctness, and (3) instruction clarity. Then we filter out examples that fail on any condition.
> * We conducted a human study to show that the LLM verifier has a high agreement with human.
> * Results show that 55.83% of the Exec-CSN examples pass all the verifications. Even if we filter out other examples, they still have a wide coverage (307 repositories) and keep the similar complexity as before.
>
> **We believe that the verification and filtering step is a kind of “specifically designed mechanisms to make sure they do not alter the semantics/functionality of the original code fragment”, and we hope adding this step to our framework can resolve your concerns about “evaluation of the individual steps”.**
>
> In the next version of the paper, we will move at least one case study example to the main context and add more case studies to the Appendix. We will also include the above quality check & filtering strategy and the human study results.
>
>
>
>
> &nbsp;
> ## W3 & Q1: Clarification of R2E’s method and Comparison to R2E
>
> We would like to first answer your question “what human effort does R2E require”. **As introduced in R2E’s paper, the authors manually resolve the error when running the setup file of some repositories**. Such errors may occur when the repos do not specify the package versions or Python versions. **If we skip this step and simply discard repositories with setup errors or functions with execution errors, the R2E framework can only produce very limited number of evaluation examples**. For instance, in the experiment in the General Response, R2E only reproduced 17 examples out of 3 repositories, while CodeBenchGen produced 196 examples out of 29 repositories.
>
> In comparison, CodeBenchGen allows the LLM to automatically resolve such errors in the execution-and-debug stage. Figure 4 shows a real example – the incorrect Python version will have the `module 'ast' has no attribute 'unparse'` error, and our method resolves it by importing from an external package: `from astunparse import unparse`.
>
>
> Here’s the summary of the comparison between R2E and CodeBenchGen (part 2 of the General Response):
> * We start with the same set of repositories (50 randomly selected licensed repositories with recent updates) and apply R2e and CodeBenchGen to produce datasets
> * R2E filters our repositories without setup files and functions without docstrings, and CodeBenchGen does not need those filterings, resulting in a larger number of seed functions.
> * CodeBenchGen is able to debug the environment errors or other execution errors in the repositories. As a result, CodeBenchGen produces a much larger final dataset (with 196 examples from 29 repositories), while R2E is only able to produce 17 examples from 3 repositories.
>
> **We hope this experiment shows that “CodeBenchGen has a higher success rate than R2E on the same set of code fragments”, as you suggested.**
>
>
>
> &nbsp;
> ## W1, 4 & 5: Experimental Details
> **We provide all the prompts here**: https://github.com/CodeBenchGen/CodeBenchGen/blob/main/resource/prompts.py. We will update them in the Appendix in the next version of the paper.
>
> We did not store the cost to build the Exec-CSN dataset. To build the above dataset with 196 examples, the cost is 1834819 input tokens + 1169062 output tokens with GPT-4o: in total $27.97 ($0.14 per example). Since Exec-CSN has 1,078 after quality filtering, the estimated cost for Exec-CSN is $0.14*1078 =  $150.92.
>
> As for the human study in Sec 4.4, we recruit 12 computer science students. Each of them is assigned 4~6 programming examples. We only have one person for each problem to get coverage across a wider range of problems. When recruiting the students, we ask for their background and only invite students who major in computer science and have at least a basic level of proficiency with programming in Python. We will include the details in the next version of the paper.

---

### Official Review · Reviewer_DbPU · 2024-11-04

**Soundness:** 2
**Presentation:** 3
**Contribution:** 2
**Rating:** 5
**Confidence:** 4

**Summary:**

The paper proposes a framework, CODEBENCHGEN, to create a scalable execution-based code generation benchmark, Exec-CSN, automatically. The authors use LLM to sandbox code and generate test cases and natural language instructions. The benchmark is diverse and has different difficulty levels. After creating the benchmark, the author evaluates the performance of multiple LLMs on this benchmark and analyzes the performance of both humans and models.

**Strengths:**

1.	The Exec-CSN benchmark demonstrates notable diversity, featuring various repositories, topics, and contributors.
2.	The evaluated LLMs exhibit considerable diversity, and it is interesting that human participation was included in the evaluation process.
3.	This framework can automatically generate code generation benchmarks and has scalability.

**Weaknesses:**

1. The modified code, test cases, and instructions are all generated by LLMs, meaning errors may occur at every step.
2. Whether the natural language instructions generated by the LLMs and code correspond without verification.
3. Although the generated test cases achieve high-line coverage, they do not guarantee complete verification of the code's correctness, especially when generated solely by the LLMs.
4. Filtering code that the model cannot generate environment and so on during the generation process raises concerns that more challenging code issues, which may require resolution, could be overlooked.

**Questions:**

1. Do the generated natural language instructions correspond to the target code, and how can the quality of the generated natural language instructions be assessed?
2. Since the model can generate test cases, why not directly input these into the model to predict the output rather than having the model output code? In other words, can the accuracy of the code generated by the model be effectively evaluated using only the test cases generated by the LLMs?
3. Is the low accuracy of the code completed by humans on the first attempt related to the lack of clarity in the descriptions of the natural language instructions?
4. What are the main sources of errors for both humans and the evaluated models?

**Details Of Ethics Concerns:**

The authors must ensure that the open-source data does not contain sensitive data, such as personal privacy information.

---

> ### Author Response · Authors · 2024-11-27
>
> Thanks for your feedback. **We take your advice and add a verification step in our framework (see part 1 of our General Response for more details)**. We also provide the answer to your other questions.
>
> &nbsp;
> ## W1-3 & Q1: Verification of evaluation examples’s quality
> As described in part 1 of the general response:
> * We use AST-based comparison and LLMs (GPT-4o) to check (1) the functionality equivalence with the seed code, (2) test correctness, and (3) instruction clarity. Then we filter out examples that fail on any condition.
> * We conducted a human study to show that the LLM verifier has a high agreement with human.
> * Results show that 55.83% of the Exec-CSN examples pass all the verifications. Even if we filter out other examples, they still have a wide coverage (307 repositories) and keep the similar complexity as before.
>
> **We hope adding this verification and filtering step to our framework can resolve your concerns that the functionality equivalence, test correctness, and instruction quality lack verification.**
>
> &nbsp;
> ## W4: Analysis of final filtering
> We agree that the filtering step can filter out some functions that are challenging to generate. However, we would like to emphasize that **our method is still able to construct some challenging examples**. For instance, after filtering, the longest target function contains 507 tokens. As shown in our analysis in Figure 8, examples with longer targets are generally more challenging to the models. We also have examples with multiple package imports (the highest one imports 8 packages).
>
> &nbsp;
> ## Q2: Test generation vs. code generation
> When we generate test cases, we provide the LLM with the ground truth implementation of the function we provide the prompt here: https://github.com/CodeBenchGen/CodeBenchGen/blob/main/resource/prompts.py). Specifically, we only ask the LLM to compare whether the generated function has the same behavior as the ground truth function. As a result, **the LLM does not need to predict the output of the function given the input**, making it easier than code generation, where the LLM does not have access to the ground truth implementation.
>
> &nbsp;
> ## Q3 & 4: Failure analysis of human programmers
> We manually checked the human study examples, and **we found multiple reasons why humans fail on the first submission**. For instance, humans could be unfamiliar with the task (e.g., execute code, read html data) or the usage of some libraries (e.g., `request`). Furthermore, we spot cases where human make careless mistakes in their first submission (e.g., in one example, the programmer misspelled a variable name). Such careless mistakes are rarely seen in LLM behaviors.
>
> We agree that “unclear instruction” could be another reason why humans fail in the first submission. In our human study (Sec. 4.4), around 17.2% are labeled as “not clear enough or not aligned well with the functionality” (the percentage is reduced to 10.2% after we apply the LLM-based verifier). We agree that this is a limitation of our framework. In fact, even human-written instructions could have clarity issues (as mentioned in this issue: https://github.com/princeton-nlp/SWE-bench/issues/72). Similar to SWE-Bench-Verified (https://openai.com/index/introducing-swe-bench-verified), **it is also possible to create a subset for Exec-CSN with only problems that are solvable by human**.

---

### Official Review · Reviewer_NUpm · 2024-11-04

**Soundness:** 2
**Presentation:** 3
**Contribution:** 2
**Rating:** 5
**Confidence:** 4

**Summary:**

The paper presents CodeBenchGen, a framework that constructs scalable, execution-based benchmarks for code generation models by adapting arbitrary code from GitHub into evaluation examples. Unlike traditional benchmarks limited to code with human-written tests or requiring substantial manual setup, CodeBenchGen uses a large language model (LLM) to automate the sandboxing, test generation, and debugging processes for code fragments, enabling their execution in isolated environments. The framework was used to create Exec-CSN, a dataset of 1,931 examples spanning 293 libraries from diverse GitHub repositories. Human and model experiments on Exec-CSN indicate that, despite improvements, existing models, including GPT-4, perform inconsistently, especially with complex code or external libraries. Results highlight challenges for code generation models and suggest that further advancements may require more diverse library coverage and improved state-tracking capabilities.

**Strengths:**

+ CodeBenchGen provides a new benchmark with full excitability and repository-level context.
+ The evaluation studied the quality of the samples from multiple perspectives and evaluated with LLMs of varied sizes

**Weaknesses:**

__Missing Most Recent Baselines__

As a new code evaluation benchmark, we hope it will be difficult enough to challenge the state-of-the-art (SOTA) LLMs so that it can help SOTA models keep exposing weaknesses and improving. Therefore, it is very important to keep the results up-to-date since, nowadays, SOTA models evolve fast, and certain capabilities can be improved drastically within one single release. Though the paper has studied a few proprietary models, but to illustrate the benchmark's practical value, I strongly encourage the authors to extend the experiments with the same metrics in Table-4 to the most up-to-date models, at least those released before ICLR submission deadline (Oct 1, 2024): GPT-4o, GPT-4o-mini, Claude-3.5-Sonnet, and Gemini-1.5 family. This is very important to maintain the dataset quality in the near future until it is included in ICLR coming next year.

__Concerns Regarding Synthetic Samples as Evaluation__

Ideally, as the code evaluation benchmark, we might not want to always ``evaluate" the model's performance on its own generated data (training on such data is fine, imo). I agree that the samples generated in this work might NOT be completely in the LLM's distribution, since they are generated based on realistic code as a prior condition, i.e., the prompt, and the data-generation instruction might interfere with the distribution a bit as well. However, it is not clear how close

I appreciate the authors' efforts in trying to study the realism and complexity using text and AST-based metrics in Section 4.2&4.3. However, the similarity in text or AST does not necessarily indicate the generated samples are as realistic as the human-written code -- the model could exploit those subtle differences that are more aligned with its own distribution, which makes the sample easier to be predicted during the evaluation, while maintaining a decent similarity. Therefore, I would encourage more experiments on (1) functional equivalence and (2) difficulty equivalence between the seed code (i.e., the code used to motivate the model's synthesis) and the synthesized code.

For functional equivalence, one quick experiment I could think of is to select a subset of realistic code snippets that are executable by the generated tests so that we can directly validate their functional equivalence, i.e., whether seed and synthetic code will reveal the same program behavior and return the same output for the same tests. The metric to report will be how often the synthesized code is functionally equivalent to the seed code. For difficulty equivalence, we could use the same subset of (seed code, synthetic code) pairs and evaluate LLMs performance on both subsets to see whether their accuracies are comparable. The metric for this experiment will be similar to Table-4 for comparison.

**Questions:**

Please address the two weaknesses mentioned in the "Weaknesses" Section.

---

> ### Author Response · Authors · 2024-11-27
>
> Thanks for your comments! Regarding your concern on functionality equivalence, **we added a quality check step as the final step of our framework** (see part 1 in the General Response for details). We also promise to add more recent models in the next version of the paper.
>
> &nbsp;
> ## Q1: More recent models
> We evaluate the performance of GPT-4o on Exec-CSN. We will include other models in the next version of the paper.
>
> | Model Name | Pass@1 | Pass@2 | Pass@5 | Pass@10 |
> |------------|--------|--------|--------|---------|
> | GPT-4o     | 37.23  | 39.45  | 41.81  | 43.34   |
>
>
> &nbsp;
> ## Q2 & 3: Example quality verification and the discussion of LLM’s distribution
>
> Here’s the summary of the part 1 of our General Response:
> * We use AST-based comparison and LLMs (GPT-4o) to check (1) the functionality equivalence with the seed code, (2) test correctness, and (3) instruction clarity. Then we filter out examples that fail on any condition.
> * We conducted a human study to show that the LLM verifier has a high agreement with human.
> * Results show that 55.83% of the Exec-CSN examples pass all the verifications. Even if we filter out other examples, they still have a wide coverage (307 repositories) and keep the similar complexity as before.
>
> Regarding your question on whether the generated target function is in the distribution of the LLM, since most functions keep the original functionality as the seed function (after filtering), we can conclude that our dataset is closer to the distribution of realistic GitHub code than affected by LLM’s distribution.

---

### Author Response · Authors · 2024-11-27
**General Response**

General Response

We thank all the reviewers for their valuable feedback. We aim to (1) resolve your concerns about the quality of the examples and (2) answer the question about the comparison to existing benchmarks as follows:
1. **We added a quality verification step in our framework** using both AST-based comparison and LLM-based verifier to check (a) the functionality equivalence with the seed code, (b) test correctness, and (b) instruction clarity. **We filter out examples that fail on any check**. We conducted a human study to show that **LLM-based verifier has a high agreement with human programmers**. We further provide statistics to show that the remaining examples still have high complexity and large repository coverage.
2. Starting from the same set of repositories (50 randomly selected licensed repositories with recent updates), **we construct new datasets with R2E and our method**. **We show that our method is able to produce much more examples than R2E** (196 examples from 29 repositories vs 17 examples from 3 repositories), even after the above quality filtering strategy.

Here are the detailed results:

&nbsp;
## 1. Quality verification and filtering

&nbsp;
### **Functionality check with AST and LLM**
We first compare the AST between the revised function and the seed function. In `421/1931 = 21.80%` of the examples, the ASTs are exactly the same. In other words, the implementations are exactly the same as the seed function.

Since there are cases where the functionality is the same, but the implementation could still be different, we further prompt gpt-4o to check the remaining cases wrt whether the functionality of the revised function is the same as before. Results show that `1178/1931 = 61.00%` of the revised functions either have the same AST as the seed function or pass the LLM-based functionality check. We will then show that this LLM-based verification is accurate using human annotations. As introduced before, we can build a “verified” subset of Exec-CSN by filtering out the examples that do not pass the functionality check.


&nbsp;
### **Test correctness check and instruction clarity check**
Similarly, we prompt gpt-4o to check whether the tests are correct and whether the instructions are clear and align well with the function. Here are the results:
```
Same or Minor differences in functionality: 1178/1931 = 61.00%
Correct & Non-trivial tests: 1719/1931 = 89.02%
Clear & Well-aligned instructions: 1841/1931 = 95.34%
Pass all 3 checks: 1078/1931 = 55.83%
```
Noted that our human study (Sec. 4.4) already verifies that 85% of the test cases are “Reasonable and not too trivial”, and 83% of the instructions are clear and well aligned with the focal method. Results in Appendix A.3. (Table 7) also show that the line coverage of the test cases is on average 95.76%.

We provide the prompts of all LLM verifiers here: https://github.com/CodeBenchGen/CodeBenchGen/blob/main/resource/llm_verifier_prompts.py


&nbsp;
### **Agreement between LLM verifier and human**
We conduct a human study to check the agreement between human and the LLM functionality verifier. Specifically, we have three programmers to label the same 20 randomly sampled examples by answering the question: whether the functionality is (a) the same, (b) different from, or (c) only has minor differences with the seed function. By “minor difference”, we only allow (1) sanity checks of the input arguments, (2) default arguments of function calls, such as revising `run(command, capture_output=True)` to `run(command, shell=True, text=True)`, and (3) difference in print statements or logging information. We also ask the annotators to specify the type of difference.

We provide these instructions and data: https://github.com/CodeBenchGen/CodeBenchGen/tree/main/resource/human_study

Here are the results:
| GPT-4o | Human  | Count |
|--------|--------|-------|
| yes    | yes    | 11    |
| yes    | minor  | 2     |
| yes    | no     | 0     |
| no     | yes    | 0     |
| no     | minor  | 1     |
| no     | no     | 6     |

In all the cases where the LLM predicts “same functionality”, the majority vote of three humans’ labels is either “same functionality” or “minor difference”. If we combine “yes” and “minor” into the same class, the Fleiss Kappa between the 3 human annotators is 0.9179.



&nbsp;
### **Statistics of the remaining examples**
The remaining examples cover 307 repositories (the seed functions cover 408 repositories), with 307/408=75.2% repository coverage rate.
The average number of code tokens for the whole script is 495.93 (before filtering: 491.88), and the number for the function to generate is 87.98 (before filtering: 86.92). This shows that the remaining 1078 examples still have high coverage and have similar complexities as before.

---

> ### Author Response · Authors · 2024-11-27
> **(cont'd) General Response**
>
> &nbsp;
> ## 2. Comparison to R2E
>
> In short, we apply R2E and our method to build datasets from the same set of repositories and show that **our method has a much higher success rate than R2E**.
>
> &nbsp;
> ### **Input of R2E/CodeBenchGen**
> Since one motivation for scalable benchmark construction methods is to create up-to-date datasets, we randomly sample 50 GitHub repositories that have recent updates (i.e., after 2024/10/1) and have the MIT License, and extract the functions and their dependencies from the repositories. To be fair, we use R2E’s function/dependency extraction code for both methods.
>
> R2E requires the repository to have a setup file and requires the functions to have docstrings.
>
> As stated in R2E’s paper, they filter out repositories without a setup file and filter out functions without docstrings. After these filters, it only has **50 functions from 7 repositories (out of 50 original repositories)** as the input to their pipeline.
>
> Our method does not apply any repo-level filtering methods and does not need to filter out functions without docstrings because there is an instruction generation step in our framework. We finally obtain **6072 functions from 36 repositories**. To reduce computation time, we randomly sample 10 functions from each repository, resulting in **297 functions from 36 repositories (out of 50 original repositories)**.
>
> &nbsp;
> ### **Resulting datasets: CodeBenchGen produces much more examples than R2E**
> We apply both R2E and our framework to construct the final dataset. To have a fair comparison, we allow 3 rounds of debugging for both methods.
>
> After applying R2E’s framework, it successfully creates tests for only **17/50 functions from 3 repositories** (by “successful”, we mean the ground truth solution can pass all test cases).
>
> In comparison, after the quality check and filtering mentioned earlier, CodeBenchGen produces **196/297 examples from 29 repositories**. We put the resulting dataset here, if you’re interested: https://github.com/CodeBenchGen/CodeBenchGen/blob/main/resource/random50_checked.json

---

### Meta-Review · Area_Chair_Ux1N · 2024-12-21

**Metareview:**

This paper introduces the CODEBENCHGEN framework to create a scalable execution-based code generation benchmark. While the reviewers acknowledge the potential and interesting aspects of this work, they have highlighted several areas requiring significant improvement. Specifically, the technical pipeline lacks sufficient detail and clarity, and the comparisons to existing benchmarks are neither rigorous nor comprehensive. Furthermore, the evaluation methodology needs to be more robust to support the claims made in the paper. The authors are encouraged to refine the technical aspects, broaden the comparative analysis, and enhance the evaluation rigor before resubmitting. With these improvements, this work could make a meaningful contribution in the future.

**Additional Comments On Reviewer Discussion:**

The reviewers have raised questions regarding the technical pipeline, rigor, comparisons to related benchmarks, and evaluation. While some concerns have been addressed, others remain unresolved.

---

### Decision · Program_Chairs · 2025-01-22

Reject